# Isotopic Signatures of Major Methane Sources in the Coal Seam Gas Fields and Adjacent Agricultural Districts, Queensland, Australia

Xinyi Lu[1], Stephen J. Harris[1], Rebecca E. Fisher[2], James L. France[2,3], Euan G. Nisbet[2], David Lowry[2], Thomas Röckmann[4], Carina van der Veen[4], Malika Menoud[4], Stefan Schwietzke[5], Bryce F. J. Kelly[1]

[1]School of Biological, Earth and Environmental Sciences, The University of New South Wales (UNSW Sydney), NSW, 2052, Australia
[2]Department of Earth Sciences, Royal Holloway, University of London, Egham, TW20 0EX, UK
[3]British Antarctic Survey, Cambridge, CB3 0ET, UK
[4]Institute for Marine and Atmospheric Research, Faculty of Science, Utrecht University, Utrecht, 3584 CC, the Netherlands
[5]Environmental Defense Fund, Berlin, Germany

*Correspondence to: Bryce F. J. Kelly (bryce.kelly@unsw.edu.au)*

**Abstract.** In regions where there are multiple sources of methane ($CH_4$) in close proximity, it can be difficult to apportion the $CH_4$ measured in the atmosphere to the appropriate sources. In the Surat Basin, Queensland, Australia, coal seam gas (CSG) developments are surrounded by cattle feedlots, grazing cattle, piggeries, coal mines, urban centres and natural sources of $CH_4$.

The characterisation of carbon ($\delta^{13}C$) and hydrogen ($\delta D$) stable isotopic composition of $CH_4$ can help distinguish between specific emissions of $CH_4$. However, in Australia there is a paucity of data on the various isotopic signatures of the different source types. This research examines whether dual isotopic signatures of $CH_4$ can be used to distinguish between sources of $CH_4$ in the Surat Basin. We also highlight the benefits of sampling at nighttime. During two campaigns in 2018 and 2019, a mobile $CH_4$ monitoring system was used to detect $CH_4$ plumes. Sixteen plumes immediately downwind from known $CH_4$

sources (or individual facilities) were sampled and analysed for their $CH_4$ mole fraction and $\delta^{13}C_{CH4}$ and $\delta D_{CH4}$ signatures. The isotopic signatures of the $CH_4$ sources were determined using the Keeling plot method. These new source signatures were then compared to values documented in reports and peer-reviewed journal articles. In the Surat Basin, CSG sources have $\delta^{13}C_{CH4}$ signatures between −55.6 ‰ and −50.9 ‰ and $\delta D_{CH4}$ signatures between −207.1 ‰ and −193.8 ‰. Emissions from an open-cut coal mine have $\delta^{13}C_{CH4}$ and $\delta D_{CH4}$ signatures of −60.0 ± 0.6 ‰ and −209.7 ± 1.8 ‰ respectively. Emissions from two

ground seeps (abandoned coal exploration wells) have $\delta^{13}C_{CH4}$ signatures of −59.9 ± 0.3 ‰ and −60.5 ± 0.2 ‰ and $\delta D_{CH4}$ signatures of −185.0 ± 3.1 ‰ and −190.2 ± 1.4 ‰. A river seep had a $\delta^{13}C_{CH4}$ signature of −61.2 ± 1.4 ‰ and a $\delta D_{CH4}$ signature of −225.1 ± 2.9 ‰. Three dominant agricultural sources were analysed. The $\delta^{13}C_{CH4}$ and $\delta D_{CH4}$ signatures of a cattle feedlot are −62.9 ± 1.3 ‰ and −310.5 ± 4.6 ‰ respectively, grazing (pasture) cattle have $\delta^{13}C_{CH4}$ and $\delta D_{CH4}$ signatures of −59.7 ± 1.0 ‰ and −290.5 ± 3.1 ‰ respectively, and a piggery sampled had $\delta^{13}C_{CH4}$ and $\delta D_{CH4}$ signatures of −47.6 ± 0.2 ‰ and −300.1 ±

2.6 ‰ respectively, which reflects emissions from animal waste. An export abattoir (meat works and processing) had $\delta^{13}C_{CH4}$ and $\delta D_{CH4}$ signatures of −44.5 ± 0.2 ‰ and −314.6 ± 1.8 ‰ respectively. A plume from a waste-water treatment plant had $\delta^{13}C_{CH4}$ and $\delta D_{CH4}$ signatures of −47.6 ± 0.2 ‰ and −177.3 ± 2.3 ‰ respectively. In the Surat Basin, source attribution is

possible when both $\delta^{13}C_{CH4}$ and $\delta D_{CH4}$ are measured for the key categories of CSG, cattle, waste from feedlots and piggeries, and water treatment plants. Under most field situations using $\delta^{13}C_{CH4}$ alone will not enable clear source attribution. It is common in the Surat Basin for CSG and feedlot facilities to be co-located. Measurement of both $\delta^{13}C_{CH4}$ and $\delta D_{CH4}$ will assist in source apportionment where the plumes from two such sources are mixed.

## 1 Introduction

If we are to achieve the goals of limiting the rise in global temperature to 2 °C as outlined in the 2015 Paris agreement of the UN Framework Convention on Climate Change (UNFCCC), we need to locate and mitigate sources of greenhouse gases due to anthropogenic industrial and agricultural activities (e.g., Ganesan et al., 2019; Pachauri et al., 2014; Nisbet et al., 2020). From measurements of the mole fraction of a gas in the atmosphere it is not always possible to isolate the source of the emission, especially if many sources are juxtaposed. However, many sources of greenhouse gases have a characteristic isotopic signature, which can be used for source attribution when used in conjunction with other data. While ethane measurements have been used previously to distinguish methane ($CH_4$) plumes from oil and gas activities vs. agricultural and other sources (e.g., Maazallahi et al., 2020; Mielke-Maday et al., 2019; Smith et al., 2015), the low ethane content in Australian coal seam gas (CSG) (Hamilton et al., 2012; Sherwood et al., 2017) renders the use of ethane measurements for source attribution impractical. This research sought to characterise isotopic signatures and to discriminate sources of $CH_4$ in the Surat Basin, from both individual sources and facilities (hereafter referred to simply as a source). The study focuses on the Surat Basin, Australia, where one of the world's largest CSG fields is co-located with large scale cattle feedlots. The gas fields are also surrounded by grazing cattle, piggeries, coal mines, urban centres and some natural sources of $CH_4$. In such regions it is a necessary but difficult task to determine how much $CH_4$ each sector contributes (Kille et al., 2019; Luhar et al., 2020; Mielke–Maday et al., 2019; Smith et al., 2015; Townsend–Small et al., 2015, 2016).

$CH_4$ is recognised as the second most abundant anthropogenic greenhouse gas species (Allen et al., 2018), contributing at least 25 % of the anthropogenic radiative forcing of warming agents (including its indirect effects) throughout the preindustrial era (Myhre et al., 2013). Counting both its radiative forcing and its wider impacts, $CH_4$ has a global warming potential 28 to 34 times higher than carbon dioxide ($CO_2$) over a 100 year time span, while on a 20 year timeline $CH_4$ is 84 to 86 times higher than $CO_2$ (Myhre et al. 2013; Etminan et al., 2016). $CH_4$ has a lifetime of about 9 years in the atmosphere compared to $CO_2$, which once added to the atmosphere takes 300 to 1000 years to be cycled out of the atmosphere (Dlugokencky et al., 2011; Joos et al., 2013; Nisbet et al., 2016). For this reason, identifying and mitigating $CH_4$ emission provides a unique opportunity to rapidly reduce the radiative forcing of the atmosphere. The atmospheric $CH_4$ mole fraction has increased by 160 % since industrialisation. The rate of increase is typically 0.4 to 14.7 ppb per year, although there was a short pause in the growth rate of atmospheric $CH_4$ between 1999 and 2006 (Dlugokencky, 2021; Schaefer et al., 2016). Since 2007, globally there has been an unremitting rise in the atmospheric $CH_4$ mole fraction with a further increase in the rate of growth noticeable after 2014

(Nisbet et al., 2014, 2019, 2020; Saunois et al., 2016). There is considerable debate about why $CH_4$ is increasing in the atmosphere, about how this $CH_4$ is apportioned between natural and anthropogenic sources, and within anthropogenic sources apportionment between agriculture versus fossil fuels (Bousquet et al., 2006; Chandra et al., 2021; Jackson et al., 2020; Kirschke et al., 2013; Nisbet et al., 2014, 2016, 2019; Rice et al., 2016; Rigby et al., 2017; Schwietzke et al., 2016; Turner et al., 2017; Worden et al., 2017). Recent ice core gas analyses of $^{14}C_{CH4}$ indicate that anthropogenic fossil fuel $CH_4$ emissions

may have been underestimated by $\sim$ 38 Tg to 58 Tg $CH_4$ per year, equivalent to $\sim$ 25 % to 40 % of recent estimates (Hmiel et al., 2020), although this result contradicts emission estimates on the size of geological fossil fuel $CH_4$ sources (Etiope et al., 2019). Gas production has continuously increased every decade over the past century, and in the last decade gas production from both conventional and unconventional (shale gas, tight gas, CSG) fields has increased by more than 30 % (BP, 2019). Particularly, unconventional gas is predicted to continue rising until the mid-century (DNV GL, 2019). The rapid expansion

of unconventional production (EIA, 2016; IEA, 2019; McGlade et al., 2013; Towler et al., 2016) is significantly increasing $CH_4$ emissions (Lan et al., 2019). It is estimated that around 14 % of total fossil fuel $CH_4$ emissions are from unconventional sources in 2020 (IEA, 2021). Thus, there is considerable interest in better quantifying $CH_4$ emissions from the gas sector.

In the Australian Government National Inventory reporting for various UNFCCC classifications, conventional gas data are

combined with the unconventional gas (CSG) data, and for some categories the sub-category details are not public. For the state of Queensland, the total UNFCCC $CH_4$ emissions reported were 1.7 Tg of which the Oil and Natural Gas sector (1.B.2) contributed 0.16 Tg (mostly from natural gas production). This is less than the total emissions from cattle (3.A.1), which contributed 0.6 Tg (Australian Government, 2019).

Various $CH_4$ surveys using a vehicle mounted analyser have been undertaken in the Surat Basin (Day et al., 2015; Hatch et al., 2018; Iverach et al., 2015; Kelly et al., 2015; Maher et al., 2014; Nisbet et al., 2020; Tsai et al., 2017). Maher et al. (2014) measured $CH_4$ mole fraction and stable carbon isotopic composition in the Tara region in 2012. Although elevated $CH_4$ mole fractions were detected within the CSG production field, no attempt was made by Maher et al. (2014) to pinpoint specific sources that caused the $CH_4$ enhancement. Several other mobile $CH_4$ surveys by Day et al. (2015), Iverach et al. (2015), Kelly

et al. (2015) and Nisbet et al. (2020) have reported high mole fractions of $CH_4$ measured from cattle feedlots, CSG co-produced water storage, ground seeps (abandoned exploration wells) and the Condamine River. Day et al. (2014) used a vehicle mounted $CH_4$ analyser to estimate $CH_4$ emissions from 37 well pads in Queensland (mostly from the Surat Basin) via a plume dispersion method. By performing traverses across the plume, and examining facilities using a probe attached to a $CH_4$ analyser, Day et al. (2014) were able to isolate and quantify emissions from well-heads, vents, pneumatic device operation and engine exhaust.

The mean emission rate from well pads was approximately 0.2 kg h$^{-1}$. In 2015, Tsai et al. (2017) surveyed a total of 137 well pads in the Surat Basin CSG field to identify and quantify $CH_4$ emissions. Their results show that emissions from all investigated well pads are between 0.008 kg h$^{-1}$ and 0.4 kg h$^{-1}$, indicating small individual site-level emissions compared with previous studies (Brandt et al., 2016). Hatch et al. (2018) also conducted mobile $CH_4$ surveys north of Tara in the Surat Basin.

Measurements of high CH$_4$ mole fraction were recorded in the region north of Dalby, but only a listing of potential sources was provided, including natural gas seeps within the Condamine River, ground seeps (abandoned gas exploration wells / uncapped water bores) or cattle feedlots. With regard to the CSG field, elevated CH$_4$ mole fractions were measured but further work was suggested to identify and separate the sources in this multi-source region. Iverach et al. (2015) and Nisbet et al. (2020) present data showing that there are substantial CH$_4$ emissions from the produced-water holding ponds (also called raw water ponds), and Nisbet et al. (2020) discuss the substantial CH$_4$ emissions from abattoirs in the Surat Basin. None of these past mobile CH$_4$ studies quantified the flux from the CSG ponds or cattle.

In the Surat Basin cattle feedlots are often located near CSG facilities as many of the feedlots are using the CSG-produced water as the water supply for the cattle. This makes it difficult to apportion the source of elevated CH$_4$ in the atmosphere from measuring CH$_4$ mole fraction alone. This is especially the case when measurements are not recorded close to the source, but rather from a distance, e.g., using an aerial survey. To distinguish CH$_4$ sources under such conditions, several studies have made use of proxy tracers such as ethane (C$_2$H$_6$), because it is often co-emitted in fossil fuel emissions (Conley et al., 2016; Dlugokencky et al., 2011; Lowry et al., 2020; Smith et al., 2015). However, the low C$_2$H$_6$ content of the gas in the Surat Basin (<1 %; Hamilton et al., 2012) limits the usefulness of this tracer. Alternatively, the isotope composition of CH$_4$ ($\delta^{13}$C$_{CH4}$ and $\delta$D$_{CH4}$) can be used to assist with identifying the source of CH$_4$, especially when used in conjunction with atmospheric and geolocation information (Fries et al., 2018; Townsend–Small et al., 2016). Each source type of CH$_4$ has a representative stable isotope ratio due to different generating processes: CH$_4$ from microbial sources is generally depleted in both $\delta^{13}$C$_{CH4}$ ( $\approx$ −62 ‰) and $\delta$D$_{CH4}$ ( $\approx$ −317 ‰) compared to thermogenic CH$_4$ from fossil fuel ($\delta^{13}$C$_{CH4} \approx$ −45 ‰, $\delta$D$_{CH4} \approx$ −197 ‰) and CH$_4$ derived from incomplete combustion (pyrogenic CH$_4$) ($\delta^{13}$C$_{CH4} \approx$ −26 ‰, $\delta$D$_{CH4} \approx$ −211 ‰) (Sherwood et al., 2017). Within these categories there is geographic variability in isotopic signature, caused by, for example the C3:C4 content of ruminant diets or combusted biomass (Brownlow et al., 2017; Fisher et al., 2017).

Isotope mixing models can be used for both regional and global scale studies to provide strong constraints on sources and sinks (Beck et al., 2012; Fisher et al., 2017; France et al., 2016; Lowry et al., 2020; McNorton et al., 2018; Nisbet et al., 2016, 2019; Rice et al., 2016; Rigby et al., 2017; Röckmann et al., 2016; Schwietzke et al., 2014, 2016; Tarasova et al., 2006). However, there is a wide range of reported CH$_4$ isotopic signatures (Sherwood et al., 2017). It is therefore important to establish suitable source signatures for the sources of interest at the regional scale. Sherwood et al. (2017) identified gaps in the isotopic characterisation in Australia. Whereas the isotopic composition of conventional fossil fuel sources is relatively well defined, there are few studies with isotope information of unconventional fossil fuels and even fewer for other CH$_4$ sources such as ruminants and waste. Table 1 lists literature reported isotopic signatures for typical CH$_4$ sources in Australia in addition to those listed in Sherwood et al. (2017), which illustrates the large variability in measured signatures across and within geographies.

Here we present mobile $CH_4$ surveys in the CSG fields in southeast Queensland that identify and characterise major $CH_4$ sources. Only plumes from clearly isolated sources or individual facilities were sampled as detailed below. Measurements of $\delta^{13}C_{CH4}$ and $\delta D_{CH4}$ from grab bag samples are then used to determine the source signature for the isolated source. These results improve the database on the isotopic signature of $CH_4$ sources in Australia, and in particular the Surat Basin. We also assess the usability of measuring just $\delta^{13}C_{CH4}$, or whether both $\delta^{13}C_{CH4}$ and $\delta D_{CH4}$ are needed to differentiate between sources.

**Table 1: Summary of isotopic characterisation of $CH_4$ sources in Australia from the literature (in addition to Sherwood et al. (2017)). NA: not applicable.**

| Source | $\delta^{13}C_{CH4}$ (‰) | $\delta D_{CH4}$ (‰) | Reference |
|---|---|---|---|
| Fossil fuels | | | |
| Coal: Surat Basin | −68.0 to −30.3 | NA | Pallasser and Stalker (2001) |
| Coal: Nagoorin Graben | −69.3 | −203.3 | Draper and Boreham (2006) |
| Coal: Surat Basin | −57.3 to −54.2 | −215.5 to 206.7 | Draper and Boreham (2006) |
| Coal: Bowen Basin | −51.2 to −38.6 | −212.9 to −201.0 | Draper and Boreham (2006) |
| Coal: Clarence Moreton Basin | −48.0 to −13.0 | NA | Doig and Stanmore (2012) |
| Coal: Bowen Basin | −66.1 to −55.7 | −213.0 to −223.0 | Golding et al. (2013) |
| Coal: Surat Basin | −57.0 to −44.5 | −233.0 to −209.0 | Baublys et al. (2015) |
| Coal: Surat Basin | −64.1 to −58.6 | NA | Hamilton et al. (2015) |
| Coal: Surat Basin | −50.8 | NA | Iverach et al. (2015) |
| Coal: Surat Basin | −56.9 to −50.1 | −210.1 to −216.3 | Day et al. (2015) |
| Coal: New South Wales (NSW) | −52.8 | −247.6 | Day et al. (2015) |
| Commercial NG: NSW | −39.4 | NA | Day et al. (2015) |
| Coal: Gunnedah Basin | −54.0 | NA | Day et al. (2016) |
| Coal: Sydney Basin | −76.8 to −61.7 | NA | Ginty (2016) |
| Coal: Sydney Basin | −66.4 | NA | Zazzeri et al. (2016) |
| Coal: Surat Basin | −80.0 to −49.0 | −310.0 to −196.0 | Owen et al. (2016) |
| Ruminants | | | |
| Cattle: NSW | −51.0 | NA | AGL Energy Limited (2015) |
| Cattle: Queensland | −49.0 | −341 | Day et al. (2015) |
| Cattle: NSW | −70.6 | NA | Ginty (2016) |
| Biomass burning | | | |
| Forest: NSW | −22.2 | NA | Ginty (2016) |
| Wetlands | | | |
| Estuary: NSW | −63.8 to −59.9 | NA | Maher et al. (2015) |
| Freshwater swamp: NSW | −51.2 | −258.6 | Day et al. (2015) |
| Estuary: Queensland | −70.0 to −37.5 | NA | Rosentreter et al. (2018) |
| Waste | | | |
| Landfill: NSW | −53.0 | −255.2 | Day et al. (2015) |
| Landfill: NSW | −44.0 | NA | AGL Energy Limited (2015) |
| Landfill: Queensland | −67.4 to −49.7 | −306.0 to −279.0 | Obersky et al. (2018) |
| Anaerobic digester | −49.7 | −326.2 | Day et al. (2015) |
| Termites | | | |
| Northern Territory | −88.2 to −77.6 | NA | Sugimoto et al. (1998) |

## 2 Method

### 2.1 Study area

The study area is situated in the Condamine region, southeast Surat Basin, and spans from Toowoomba, Dalby, Chinchilla, to
Miles and the surrounding area. The size of the total study area is approximately 50,000 km$^2$. Figure 1 shows potential major sources of CH$_4$ in the study area. Location and capacity data (where available) of CSG wells, petroleum pipelines, coal mines, cattle feedlots, piggeries, landfills, wastewater treatment plants (WWTP) and abattoirs (export abattoirs that include both meat works and additional processing, and smaller licensed abattoirs) were retrieved from the Queensland Government Open Data Portal (https://www.data.qld.gov.au). CSG processing facilities and raw water ponds were manually located using Google
Maps (Google LLC, USA) and Queensland Globe (Queensland Government, 2020a). The locations of ground seeps discussed are a combination of those reported in Day et al. (2015) and field measurements. In Day et al. (2015) and this study, ground seeps refer not only to natural CH$_4$ seeps but also to abandoned exploration wells.

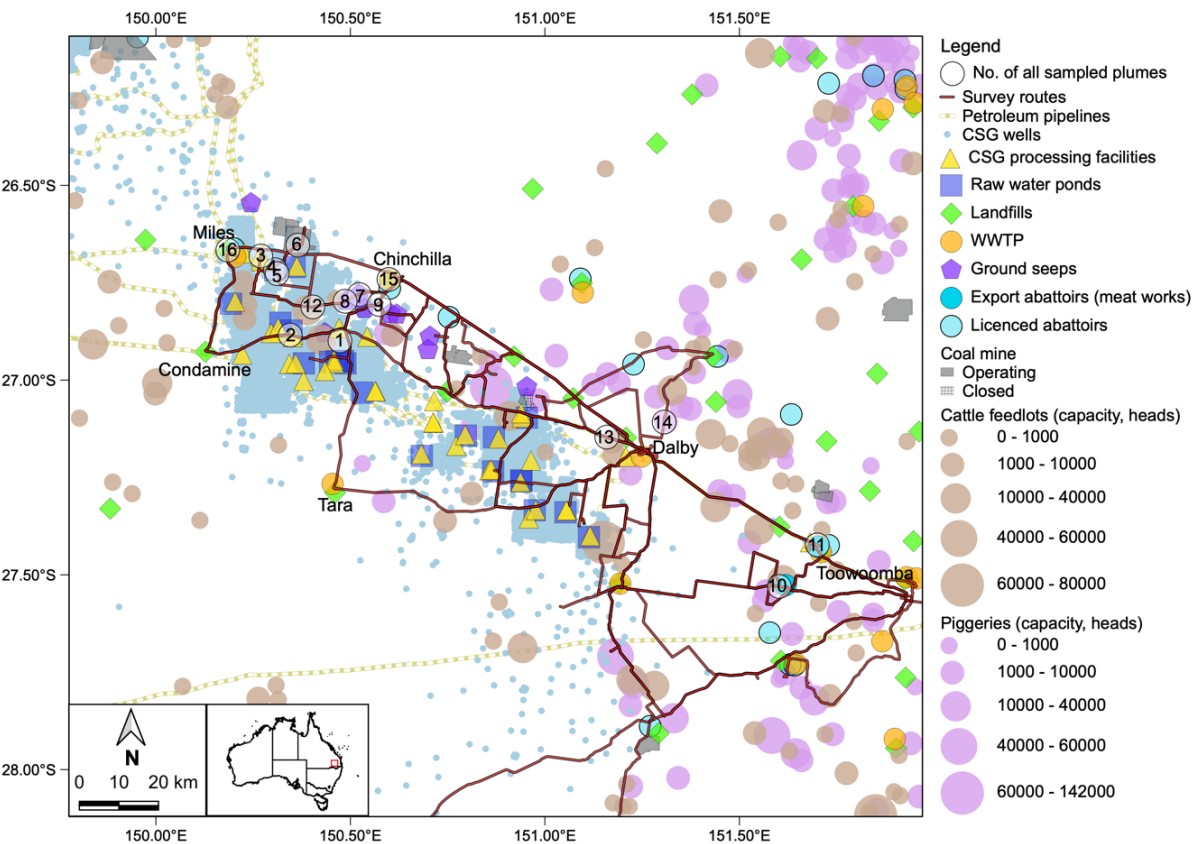

**Figure 1: Map of the study area with survey routes and potential CH$_4$ sources. Inset map shows the location in south-eastern Queensland (Inset map data: Australian Government (2020), Administrative Boundaries © Geoscape Australia). The positions of the sampled CH$_4$ plumes are numbered 1 through 16.**

The Surat Basin holds more than 60 % of Australia's total proven gas reserves (Australian Competition and Consumer Commission, 2020). The study area covers many of the intensive CSG exploration and production petroleum leases (PLs). In 2018 gas was produced from 5153 exploration, appraisal and production CSG wells, as well as a small number of oil and coal exploration wells within the region (Queensland Government, 2020b). All the CSG in the Surat Basin is produced from the Walloon Coal Measures (Queensland Government, 2020c). Within the region there are 42 processing facilities, 21 raw water

ponds, and over 2000 km of pipelines. To the east and north of the CSG region studied there are 4 operating open-cut coal mines, and one recently closed. In total, they produced 17.5 million tonnes of saleable coal from 1 July 2018 to 30 June 2019 (Queensland Government, 2019). Coal and gas fired power stations are another potential source of $CH_4$. In the study area, seven power stations (5 CSG-fired and 2 coal-fired) are operational; together they account for 0.15 % of the $CH_4$ emissions for the south-east portion of the Surat Basin CSG fields (Neininger et al., in review). $CH_4$ sources from the agricultural sector

are also considerable. Cattle and pigs are two of the most important commodities in Queensland. There are also other anthropogenic sources of $CH_4$ in the town areas, including landfills, wastewater treatment plants, domestic wood heaters, and automobiles, among others. Natural $CH_4$ seeps (the Condamine River near Chinchilla) and emissions from abandoned coal exploration wells have also been mapped within the region (Day et al., 2013, 2015; Iverach et al., 2014; Kelly et al., 2015, 2017; Kelly and Iverach, 2016).


Ruminants such as cattle produce $CH_4$ in the rumen, which is then emitted to the atmosphere. A study from the Australian Commonwealth Scientific and Industrial Research Organisation (CSIRO) reported that cattle grazing is the main contributor to the total regional $CH_4$ emissions in the Surat Basin. Two sources of community concern, CSG and feedlots, contribute less to the regional emissions than the grazing cattle (Luhar et al., 2020).


Within the Condamine Natural Resource Management Region there are ≈ 560,000 cattle (meat (feedlot and pasture) ≈ 520,000 and dairy ≈ 40,000) (Australian Bureau of Statistics, 2020). In 2018 there were 65 feedlots in the region, the largest, Grassdale Feedlot, holding up to 75,000 cattle (Beef Central, 2020; Queensland Government, 2018a). As part of this study, we sampled the plume downwind of Stanbroke feedlot (No. 12 in Fig. 1) in 2018. This feedlot has a capacity of 40,000 cattle. Most cattle

in the region are in the surrounding dryland farming districts. These cattle graze a variety of crops and native grasses (we label these grazing cattle). We sampled a plume from roadside feeding grazing cattle near Dalby in 2019 (No. 13 in Fig. 1).

Pigs produce $CH_4$ via the anaerobic degradation of organic matter by bacteria in their digestive systems. Manure in the piggeries is another source of $CH_4$ due to processing by microbial consortia (Flesch et al., 2013). Firstly, the increasing

acidogenic bacteria in the manure convert substrates into volatile fatty acids (VFAs), $CO_2$ and hydrogen [H]. The methanogenic bacteria then produce $CH_4$ from organic acids (Monteny et al., 2006). There are 67 piggeries spread throughout the Natural

Resource Management Region collectively holding ≈ 270,000 pigs in 2018−2019 (Australian Bureau of Statistics, 2020). In the region, the largest piggery holds up to 142,000 pigs (Queensland Government, 2018b). In 2019 we sampled a plume downwind of Albar Piggery (No. 14 in Fig. 1), which has a registered capacity of 4,980 pigs.


Other agriculture-related $CH_4$ emissions in the region are from urban waste biosolid and animal manure that are used to fertilise the soils in the irrigation districts and abattoirs. In Queensland there are many abattoirs that process meat for both domestic use and export. The number of abattoirs documented in the area is 20; most of these abattoirs are small (licensed abattoirs), but there are two large export abattoirs: Beef City (Abattoir A; No. 10 in Fig. 1) and Oakey Beef Exports (Abattoir B; No. 11 in Fig. 1). Beef City is one of only two comprehensive beef processing plant and feedlot operations in Australia, and one of the largest such facilities worldwide. The feedlot has a capacity of 26,500 head, and 1,134 cattle are processed in the beef processing plant per day. Oakey Beef Exports processes up to 1,200 head of cattle per day (NH Foods, 2020). Both facilities produce a range of meat and meat by-products.

Urban landfills are strong sources of atmospheric $CH_4$ (Nisbet et al., 2020). Isotopic signatures of gas emitted from landfill gas collection systems or covering soil vary depending on factors such as deposited materials, temperature, or the degree of $CH_4$ oxidation in the above soil layers (Zazzeri et al., 2015). As part of this study, we sampled the plume downwind of the Chinchilla domestic landfill (26.74°S, 150.60°E; No. 15 in Fig. 1). The landfill has a disposal area of approximately 0.07 $km^2$ for municipal waste and was closed to the public in 2014. This landfill is typical of many small-town landfills in the region, and when operational it accepted mixed dry and solid organic domestic waste, commercial and industrial waste. These landfills have a simple design and typically have a clay lining and soil cover. A full listing of the landfills in the study area and the materials deposited within each are listed in Western Downs Regional Council (2021a).

Wastewater treatment plants are another source of urban $CH_4$ emissions, and there is a treatment plant at every major town in the region. In 2019 we sampled the plume immediately downwind of the Miles wastewater treatment plant (No. 16 in Fig. 1). There, the sludge was treated in digestion tanks under anaerobic conditions. The liquid from the tanks was then transferred to the aerobic lagoons for further purifying (Western Downs Regional Council, 2021b).

Natural sources in the region include wetlands, termites, and natural fires by lightning (Lu et al., 2020). We did not attempt to characterise these natural sources as part of this study. Below we focus on the major anthropogenic sources identified in Luhar et al. (2018, 2020).

## 2.2 Mobile CH$_4$ monitoring system

To map the major CH$_4$ sources in the Surat Basin, we measured the CH$_4$ mole fraction in the atmosphere as we drove along the main roads throughout the major CSG and agricultural regions of the Surat Basin. In 2018 and 2019 over 2000 km of measurements were made using a Los Gatos Research Ultraportable Greenhouse Gas Analyser (LGR-UGGA) (model 915-0011, Los Gatos Research, Inc., USA). This instrument uses off-axis integrated cavity output spectroscopy (Baer et al., 2002), and records the CH$_4$ mole fraction every second in parts per million (ppm). The manufacturer's stated precision is 1 standard deviation of < 2 parts per billion (ppb) and a measurement range of 0 to 100 ppm. These analysers were further characterised by Allen et al. (2019). In-field calibration using southern-ocean air supplied by CSIRO is discussed further below. The air inlet was attached to a mast mounted on top of the vehicle (2.7 m above ground surface). Ambient air was then pumped into the LGR-UGGA through a Teflon tube. A Hemisphere global positioning system (GPS) (Model A326, Hemisphere GNSS, Inc., USA) was also mounted on the roof, measuring the geolocation to within 8 cm (2 standard deviations, GNSS 2017). The air inlet tube was 2.5 m long; this results in a lag between the GPS recorded time stamp and the analyser time stamp. Using standard air this was determined to be 7 s. It was not the goal of the project to do detailed plume analyses. Driving speed was not independently continuously measured, and only a lag time correction was made. As a result, the surveys were not precisely positioned. When a major plume was traversed, we returned to the centreline of the plume and remained stationary to georeference the plumes shown in Fig. 2. The car was stationary for up to half an hour while the air samples were collected. In Fig. 2 the plume positions are accurately located, but away from the plumes the survey results are only approximate to within the order of tens of meters.

For a small portion of the 2018 campaign, plume mapping was done using a Picarro G2201-i cavity ring−down spectrometer (CRDS) (Picarro, Inc., USA), due to the failure of the LGR-UGGA unit. The Picarro reported precision (1 standard deviation, 30 seconds average) of CRDS for CH$_4$ mole fraction is 5 ppb + (0.05 % of the reading) for $^{12}$C and 1 ppb + (0.05 % of the reading) for $^{13}$C in high precision (HP) mode with an operational range of 1.2 to 15 ppm. Under the same operation mode, the instrument precision (1 standard deviation, 5 minutes average) for $\delta^{13}C_{CH4}$ is < 1.15 ‰ with a maximum drift (over 24 hours) of < 1.15 ‰ at 10 ppm. Previous studies have also characterised the Picarro G2201-i performance (e.g., Assan et al., 2017; Rella et al., 2015). For the Picarro portion of the surveying we recorded the GPS location using a Kinetic Lite GPS application (Mothership Software Ltd., UK). Using the standard air, we determined the time lag between the real-time GPS location reading and the display of mole fraction reading on the Picarro G2201-i CRDS to be 3 min and 40 s. Using this timing offset, we adjusted the time stamp for the analyser data.

One-point calibrations for the two instruments were conducted before and after each survey using Southern Ocean air provided by CSIRO. The calibration gas was placed into 3 litre SKC FlexFoil PLUS sample bags (SKC Inc., USA) for shipping and analysed at the greenhouse gas laboratory of Royal Holloway, University of London (RHUL) to determine the $\delta^{13}C_{CH4}$ for the

calibration air (−47.2 ± 0.05‰). RHUL also measured the $CH_4$ mole fraction of the calibration gas (1801.2 ± 0.5 ppb). The isotope value measured by RHUL (−47.2 ± 0.05‰) also closely resembles the value from flasks (−47.2 ± 0.04‰, mean ± standard deviation for 12 flasks collected) collected at Cape Grim and measured at the Institute of Arctic and Alpine Research (INSTAAR), University of Colorado (White et al., 2018) around the same time as the Southern Ocean cylinder was filled by

CSIRO (29 June 2016 to 11 August 2016). The in-field standard deviations for mean $CH_4$ mole fraction measurements of the reference standard across all days were 4.9 ppb (2018) and 9.6 ppb (2019) for LGR-UGGA and 5.3 ppb (2018) for Picarro G2201-i CRDS. This repeatability is better than reported in Takriti et al. (2021).

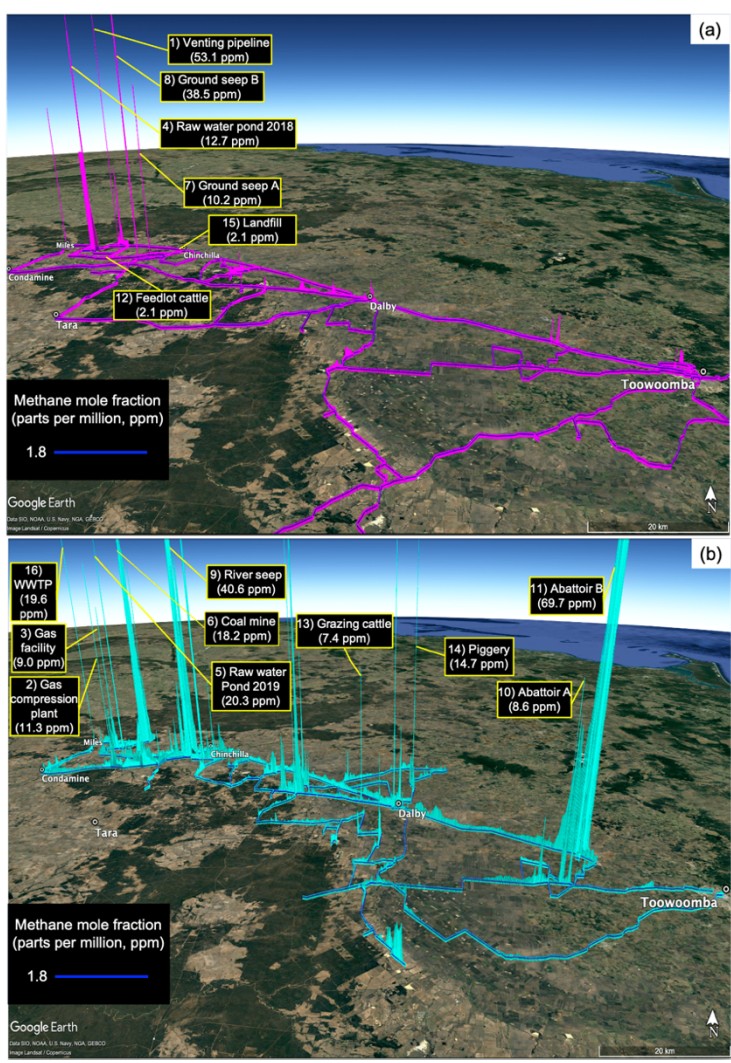


**Figure 2: The vehicle-mounted $CH_4$ survey routes throughout the Surat Basin. Daytime measurements are represented by a magenta ribbon and nighttime measurements by a cyan ribbon. A linear scale is used to represent the measured $CH_4$ mole fraction. For all sampled plumes, the highest recorded $CH_4$ mole fraction is indicated (image © Google Earth).**

## 2.3 Sampling and measurement methods

During the two campaigns in 2018 and 2019, driving speed was controlled between $10-80$ km h$^{-1}$ for surveys with LGR-UGGA and $10-40$ km h$^{-1}$ for surveys with Picarro G2201-i CRDS where traffic conditions were suitable. The lower driving speed coupled with real-time CH$_4$ mole fraction readings allowed us to detect plumes associated with potential CH$_4$ sources. When a constant plume was detected, we collected 10 air samples for isotopic analysis downwind of the plume by pumping air into 3 litre SKC FlexFoil PLUS sample bags with polypropylene fittings using a 2-litre medical syringe. In total, over 160

air samples were collected from 16 major sources in the Surat Basin CSG fields. On the day the samples were collected they were analysed for CH$_4$ mole fraction and $\delta^{13}C_{CH4}$ in the field using Picarro G2201-i CRDS for data quality-control purposes. In 2018 the root-mean-square error (RMSE) between the University of New South Wales (UNSW Sydney) Picarro 2201-i CRDS and the RHUL Picarro G1301 CRDS (detailed below) was 0.437 (ppm; Fig. A1 (a)) and in 2019 the RMSE between the UNSW Sydney Picarro 2201-i CRDS and the Institute for Marine and Atmospheric research Utrecht (IMAU) continuous-

flow isotope ratio mass spectrometry (CF-IRMS) (detailed below) was 0.232 (ppm; Fig. A1 (b)).

The sampling of plumes favours those sources that happen to be upwind and close to a public road. The objective of this study was not to quantify the emission rate (flux) of individual sources. Rather, our aim was to characterise the isotopic source signatures of potential significant sources of CH$_4$ in the region. We did not have permission to access private properties or

industrial sites, which was a significant constraint on sampling. All samples collected in this study are from publicly accessible locations. When a plume was located, we sampled several locations within the plume to maximise the range of CH$_4$ mole fraction values that could be obtained within the limits of public access. Sampling a large range of CH$_4$ mole fraction values assists with minimising the uncertainties for each source signature derived using the Keeling plot method in combination with Bayesian linear regression (see Sect. 2.4).


In 2018, air samples were analysed in the greenhouse gas laboratory at RHUL for CH$_4$ mole fraction and $\delta^{13}C_{CH4}$ using the Picarro G1301 CRDS (Picarro, Inc., USA) and modified gas chromatography isotope ratio mass spectrometry (GC-IRMS) system (Trace Gas and Isoprime mass spectrometer, Elementar UK Ltd., UK) respectively. The Picarro G1301 CRDS was calibrated to the WMO X2004A scale using NOAA (National Oceanic and Atmospheric Administration) air standards

(Dlugokencky et al., 2005; Fisher et al., 2006, 2011; WMO, 2020). For CH$_4$ mole fractions analysis, each sample was analysed for 210 seconds on the Picarro G1301 CRDS with a reproducibility of $\pm$ 0.3 ppb and the mean CH$_4$ mole fraction of the last 90 seconds of the analysis was recorded. For $\delta^{13}C_{CH4}$ analysis, samples with mole fractions above 6 ppm were diluted with zero grade nitrogen to fit the dynamic range for the GC-IRMS and then measured in triplicate on the VPDB (Vienna Pee Dee Belemnite) scale. A fourth analysis was made if the standard deviation of the first three analyses was greater than the target

instrument precision of 0.05‰. A portion of the samples (from 13 plumes) was further analysed in the IMAU for CH$_4$ mole fraction, $\delta^{13}C_{CH4}$ and $\delta D_{CH4}$ using continuous-flow isotope ratio mass spectrometry (CF-IRMS) (Thermo Finnigan Delta plus

XL, ThermoFinnigan MAT, Germany) (Brass and Röckmann, 2010; Eyer et al., 2016). All samples were measured directly with the automated extraction system. For the subsequent IRMS measurement, the $CH_4$ in the air from most bags were preconcentrated for 10 minutes at a flow rate of 6 mL min$^{-1}$ for $\delta D_{CH4}$ and 4 mL min$^{-1}$ for $\delta^{13}C_{CH4}$, but for samples reported by RHUL that had a $CH_4$ mole fraction larger than 6 ppm they were processed for a shorter time in order to extract a quantity of $CH_4$ similar to the reference air. The $CH_4$ from 60 mL of air was extracted for each $\delta D_{CH4}$ measurement, and from 40 mL for $\delta^{13}C_{CH4}$ measurements. $\delta D_{CH4}$ measurements are given on the VSMOW (Vienna Standard Mean Ocean Water) scale. A one-point calibration was done using a reference cylinder with the following assigned values $CH_4$ mole fraction: 1975.5 ppb, $\delta^{13}C_{CH4}$: −48.2 ‰ (VPDB), $\delta D_{CH4}$: −90.8 ‰ (VSMOW). In 2019, air samples were analysed at IMAU for $CH_4$ mole fraction, $\delta^{13}C_{CH4}$ and $\delta D_{CH4}$ using the same CF−IRMS as 2018. Samples with reported $CH_4$ mole fraction larger than 3 ppm by UNSW Sydney were sampled at a lower flow rate in order to extract a quantity of $CH_4$ similar to the reference air. A one-point calibration was done using a reference cylinder with the following assigned values $CH_4$ mole fraction: 1970.0 ppb, $\delta^{13}C_{CH4}$: −48.1 ‰ (VPDB), $\delta D_{CH4}$: −88.3 ‰ (VSMOW). Due to the high precision of the RHUL GC-IRMS measurements of < 0.05 ‰ for $\delta^{13}C$, the IMAU IRMS measurements of < 0.1 ‰ for $\delta^{13}C$ and < 2 ‰ for $\delta D$, reliable source signatures can usually be derived for elevations of 100–200 ppb above the background.

## 2.4 Data analysis

The $\delta^{13}C_{CH4}$ and $\delta D_{CH4}$ for $CH_4$ sources of each detected plume were determined using the Keeling plot approach (Keeling, 1958; Pataki et al., 2003) shown in Eq. (1):

$$\delta_{(a)} = \left[CH_{4(b)}\right](\delta_{(b)} - \delta_{(s)}) * 1/\left[CH_{4(a)}\right] + \delta_{(s)} \tag{1}$$

where $[CH_{4(b)}]$ and $\delta_{(b)}$ are the $CH_4$ mole fraction and $\delta^{13}C_{CH4}$ (or $\delta D_{CH4}$) of the background air, $[CH_{4(a)}]$ and $\delta_{(a)}$ are the $CH_4$ mole fraction and $\delta^{13}C_{CH4}$ (or $\delta D_{CH4}$) of the atmosphere and $\delta_{(s)}$ is the $\delta^{13}C_{CH4}$ (or $\delta D_{CH4}$) of the mean source, respectively. The intercept ($\delta_{(s)}$) of the linear regression between $\delta_{(a)}$ and $1/[CH_{4(a)}]$ represents the isotopic signature of the source mixed in the background ambient air. The Keeling plot method requires the background air $CH_4$ mole fraction and isotopic signature to be constant during the period of observation. The time it takes to collect the 10 samples is approximately 30 minutes, and normally the background air composition does not change during the period of sampling. The mobile survey readings show that the background $CH_4$ mole fraction was stable in 2018 and 2019 daytime and nighttime surveys (Fig. A2), which supports this assumption. For each Keeling data set the linear regression line and credible interval (analogous to confidence interval) were determined using the PyMC3 Bayesian regression package (Salvatier et al., 2016). The regression methodology was selected based on the fact that there are bivariant correlated errors in both the x and y variables (e.g., Miller and Tans, 2003; Zazzeri et al., 2016) and the number of samples in each plume set was small (<= 10). Bayesian regression was used since it is a robust algorithm that balances uncertainty in both the x and y axis data (Jaynes, 1999), it is suitable for small data sets (Baldwin and Larson, 2017), and it has been demonstrated to yield more reliable isotopic signatures at low mole fractions with low sample numbers (Zobitz et al., 2007).

# 3 Results and Discussion

## 3.1 Regional plume mapping and the benefits of sampling at nighttime

Two campaigns with over 2000 km routes were conducted in September 2018 and from August to September 2019 (Fig. 1). The $CH_4$ mole fraction in the atmosphere 2.7 m above the ground was mapped between Toowoomba and Miles (a distance of approximately 200 km. Surveys of $CH_4$ mole fraction during both daytime and nighttime are shown in Google Earth (Fig. 2). In 2018, we did not detect plumes from coal mines, river seeps, abattoirs, piggeries or WWTPs thus we shifted our focus from daytime surveying in 2018 to nighttime surveying in 2019. During the day the sunshine heats the ground, which warms the air immediately above the surface. This causes the plumes to rise rapidly and mix with background air within the growing boundary layer, rather than accumulating within the nocturnal boundary layer. This results in daytime plumes either being missed during the mobile surveys or having a limited range of $CH_4$ mole fraction values. By contrast, at night during weak to moderate wind conditions the plumes typically disperse slowly within the stable nocturnal boundary layer when there is a large temperature inversion (Stieger et al., 2015). This enabled us to sample isolated source plumes that have a greater spread of $CH_4$ mole fraction, which improves determination of the line of best fit in Keeling plots and minimises the uncertainties of the derived isotopic source signatures. As part of developing an inventory (Neininger et al., in review) in the region, all major $CH_4$ sources were located and were georeferenced to guide nighttime sampling. Also, most facilities were well lit, which assisted with source identification. The contrast in the magnitude of the $CH_4$ mole fraction measured in the field between the daytime and nighttime surveys is clearly visible in Fig. 2. The distribution of the $CH_4$ spikes demonstrates the complex spread of the sources in the study area. Overall, measured $CH_4$ mole fraction ranged from 1.8 to 69.7 ppm – the highest value was recorded in a plume downwind of Oakey Beef Exports (Abattoir B).

## 3.2 Source isotopic signatures

The Keeling plot results of $CH_4$ source signature calculations are listed in Table 2 and shown in Fig. 3, and the Keeling plots are shown in Figs. A3–A7 in Appendix A. For each $\delta^{13}C_{CH4}$ (‰) and $\delta D_{CH4}$ (‰) isotopic signature both the posterior standard deviation and the credible interval were determined. The variability in the credible interval is primarily due to both the sampled $CH_4$ mole fraction range and the number of data points used in the Keeling plot analysis as shown in Fig. A8.

**Table 2: CH₄ source signature results for plumes sampled in the Surat Basin 2018 and 2019 campaigns. CH₄ excess over background (ppm) for the samples that were used to calculate the source signature. δ¹³C_CH4 (‰) and δD_CH4 (‰) are reported along with the Bayesian posterior distribution mean, standard deviation and 95 % credible interval (in brackets). NA: not applicable.**

| No. | Upwind source | Sample date and time: D – daytime, N – nighttime | Location Latitude & Longitude | Wind direction | Distance from source (km) | CH₄ excess over background (ppm) | $\delta^{13}C_{CH4}$ (‰) | $\delta D_{CH4}$ (‰) | No. of samples $\delta^{13}C$ & $\delta D$ |
|---|---|---|---|---|---|---|---|---|---|
| | CSG infrastructures | | | | | | | | |
| 1 | Venting pipeline | 20/9/18, D | 26.89935° S, 150.47316° E | SW | <0.1 | 32.7 | −54.5 ± 0.1 (−54.8, −54.3) | −198.8 ± 1.0 (−200.8, −196.6) | 9 & 5 |
| 2 | Gas compression plant | 22/9/18, N | 26.88442° S, 150.34508° E | NE | 0.6 | 1.9 | −53.7 ± 0.4 (−54.5, −53.0) | −193.8 ± 2.9 (−199.6, −188.2) | 9 & 5 |
| 3 | CSG facility | 2/9/19, N | 26.68141° S, 150.26974° E | W | 0.1 | 4.7 | −55.6 ± 0.4 (−56,4, −54.7) | −207.1 ± 2.9 (−212.6, −201.2) | 6 & 6 |
| 4 | Raw water pond (2018) | 22/9/18, D | 26.71666° S, 150.30706° E | SE | 1.0 | 0.2 | −50.9 ± 2.8 (−56,6, −45.6) | NA | 7 & NA |
| 5 | Raw water pond (2019) | 1/9/19, N | 26.72668° S, 150.31171° E | NW | 1.0 | 1.5 | −51.9 ± 2.3 (−56.7, −47.2) | −195.6 ± 3.6 (−202.8, −188.7) | 3 & 3 |
| | Coal mining | | | | | | | | |
| 6 | Coal mine | 1/9/19, N | 26.65342° S, 150.36480° E | NW | 2.7 | 11.4 | −60.0 ± 0.6 (−61.1, −58.9) | −209.7 ± 1.8 (−213.6, −206.3) | 5 & 5 |
| | Ground and river seeps | | | | | | | | |
| 7 | Ground seep A | 19/9/18, D | 26.78030° S, 150.52285° E | NW | <0.1 | 4.1 | −59.9 ± 0.3 (−60.5, −59.2) | −185.0 ± 3.1 (−191.1, −178.8) | 8 & 3 |
| 8 | Ground seep B | 19/9/18, D | 26.79769° S, 150.48646° E | NW | <0.1 | 16.2 | −60.5 ± 0.2 (−60.9, −60.1) | −190.2 ± 1.4 (−192.9, −187.6) | 8 & 5 |
| 9 | River seep | 2/9/19, N | 26.80560° S, 150.57352° E | E | 0.3 | 6.5 | −61.2 ± 1.4 (−63.9, −58.4) | −225.1 ± 2.9 (−230.9, −219.3) | 4 & 4 |
| | Export abattoirs (meat works) | | | | | | | | |
| 10 | Abattoir A | 12/9/18, N | 27.52994° S, 151.60254° E | E | 1.1 | 5.2 | −46.0 ± 0.4 (−46.7, −45.3) | NA | 9 & NA |
| 11 | Abattoir B | 4/9/19, N | 27.42310° S, 151.70059° E | E | 0.2 | 4.5 | −44.5 ± 0.2 (−44.9 −44.0) | −314.6 ± 1.8 (−318.2, −311.2) | 9 & 9 |
| | Agriculture | | | | | | | | |
| 12 | Feedlot cattle | 20/9/18, D | 26.81209° S, 150.40338° E | SW | 0.1 | 0.2 | −62.9 ± 1.3 (−65.2, −60.3) | −310.5 ± 4.6 (−319.1, −301.2) | 9 & 5 |
| 13 | Grazing cattle | 29/8/19, N | 27.14643° S, 151.15916° E | NE | <0.1 | 1.3 | −59.7 ± 1.0 (−61.7, −57.5) | −290.5 ± 3.1 (−296.5, −284.3) | 6 & 6 |
| 14 | Piggery | 5/9/19, N | 27.10768° S, 151.30661° E | NE | 0.6 | 2.3 | −47.6 ± 0.2 (−48.0, −47.1) | −300.1 ± 2.6 (−304.9, −294.9) | 10 & 10 |
| | Landfill | | | | | | | | |
| 15 | Chinchilla landfill | 20/9/18, D | 26.74148° S, 150.59905° E | SW | <0.1 | 0.1 | −52.1 ± 3.6 (−59.0, −45.3) | NA | 10 & NA |
| | WWTP | | | | | | | | |
| 16 | Miles WWTP | 2/9/19, N | 26.66612° S, 150.18469° E | W | <0.1 | 6.5 | −47.6 ± 0.2 (−47.9, −47.2) | −177.3 ± 2.3 (−182.0, 173.0) | 6 & 6 |

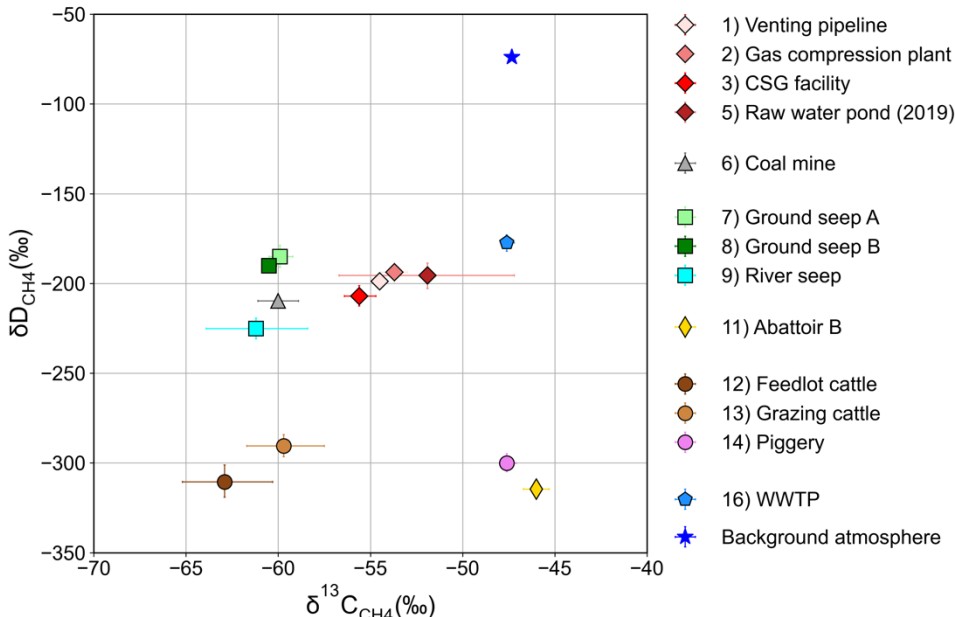

Figure 3: Dual isotope plot of all measured CH4 sources in the study. For markers with missing error bars the Bayesian credible interval were smaller than the symbol size. Please refer to Table 2 for detailed information of plotted data.

### 3.2.1 Coal seam gas infrastructures

There are many portions of the CSG production and processing lifecycle where CH4 can be released, either accidentally or by deliberate venting. CH4 can be released intentionally at high-point vents along the produced water pipelines, outgassed from raw water ponds, or released as part of other venting or flaring operations. Unintentional CH4 releases can occur anywhere where there are joints and seals, which can be at well heads, or along gas distribution lines, compression stations, and processing plants. The isotopic signatures of the resultant CH4 emissions may vary depending on the origin of the gas within a gas field. The production processes and conditions of the coal and associated groundwater are not constant throughout a region, which can result in variations of the isotopic composition of the gas both spatially and with depth (Hamilton et al., 2015; Iverach et al., 2017). In the Surat Basin CSG fields, all CH4 plumes from active CSG production and processing sources sampled show relatively little variability and sit in a distinct cluster isolated from non-CSG sources in Fig. 3. These plumes were from a range of sources including a high-point vent on a produced water pipeline, a gas compression plant, a raw water pond (measured in both 2018 and 2019 campaign), and a CSG facility (No. 1–5 in Table 2 and Figs. 1, 2 and 3).

Downwind of the high point vent on the produced water pipeline we sampled a plume with a maximum CH4 mole fraction reading of 35.0 ppm (wind direction was SW) approximately 15 m from the venting point (No. 1 in Table 2 and Figs. 1, 2 and 3). The $\delta^{13}C_{CH4}$ and $\delta D_{CH4}$ signatures of the vented gas were $-54.5 \pm 0.1$ ‰ and $-198.8 \pm 1.0$ ‰.

Another major CSG $CH_4$ plume detected was associated with nighttime operations at the APLNG Talinga gas compression plant (No. 2 in Table 2, Figs. 1, 2 and 3). On the evening of sampling, this plume extended for 17 km (see Fig. 2). The peak $CH_4$ mole fraction measured was 11.3 ppm approximately 0.6 km downwind of the facility. The sampled gas had $\delta^{13}C_{CH4}$ and $\delta D_{CH4}$ signatures of $-53.7 \pm 0.4$ ‰ and $-193.8 \pm 2.9$ ‰, respectively.

The Glen Eden raw water pond was surveyed on 22 September 2018 and 1 September 2019 (No. 4–5 in Table 2, Figs. 1, 2 and 3). This pond is one of the many in-field storages that temporarily hold water gathered from each CSG well-head (QGC, 2014). The $\delta^{13}C_{CH4}$ signatures of the gas sampled were $-50.9 \pm 2.8$ ‰ and $-51.9 \pm 2.3$ ‰ in 2018 and 2019, respectively, with a $\delta D_{CH4}$ signature of $-195.6 \pm 3.6$ ‰ in 2019. No significant differences were found between the $\delta^{13}C_{CH4}$ signatures from the two campaigns for this pond. The results are similar to those from a previous study in the area with a $\delta^{13}C_{CH4}$ signature of $-50.8$ ‰ (90 % CI, $-55.7$ ‰ to $-45.8$ ‰) from CSG water storage (Iverach et al., 2015).

In September 2019 we intersected a $CH_4$ plume emanating from a CSG gas transfer hub. The peak $CH_4$ mole fraction measured in the plume 150 m east and downwind of the facility was 7 ppm. The $\delta^{13}C_{CH4}$ and $\delta D_{CH4}$ signatures were $55.6 \pm 0.4$ ‰ and $-207.1 \pm 2.9$ ‰ respectively (No. 3 in Table 2, Figs. 1, 2 and 3).

Draper and Boreham (2006) reported that the $\delta^{13}C_{CH4}$ signature for $CH_4$ from the Surat Basin Walloon Coal Measures (WCM) ranged from $-57.3$ ‰ to $-54.2$ ‰, indicating secondary biogenic $CH_4$ with a minor thermogenic component. More recent studies by Hamilton et al. (2014, 2015) and Baublys et al. (2015) report $\delta^{13}C_{CH4}$ signature ranging from $-64.1$ ‰ to $-44.5$ ‰ with median of $-52.0$ ‰. These have a $\delta^{13}C_{CH4}$ range of approximately 20 ‰, while all above ground measurements fall within a narrower range. Iverach et al. (2015) and Day et al. (2015) reported $\delta^{13}C_{CH4}$ signatures from $-56.9$ ‰ to $-50.1$ ‰, and in this study we measured $\delta^{13}C_{CH4}$ signatures from $-55.6 \pm 0.4$ ‰ to $-50.9 \pm 2.8$ ‰ (Fig. 4). Owen et al. (2016) found that the $\delta^{13}C_{CH4}$ values for the gas reservoir (200–500 m) for coal measures in the Surat Basin were between $-58.0$ ‰ and $-49.0$ ‰. This is consistent with our study as the commercially produced gas is extracted from coal seams at depths >200 m (Queensland Government, 2020b).

The $\delta D_{CH4}$ data for the WCM in the Surat Basin are relatively sparse in the literature. Early studies of the Surat Basin CSG found a range of $\delta D_{CH4}$ signatures from $-215.5$ ‰ to $-206.7$ ‰ (Draper and Boreham, 2006). Baublys et al. (2015) and Day et al. (2015) reported that gas from the WCM in the same area had values from $-233.0$ ‰ to $-209.0$ ‰ and from $-216.3$ ‰ to $-210.1$ ‰. In general, the determined $\delta D_{CH4}$ signatures (median = $-197.2$ ‰) of gas from CSG infrastructures in this study are approximately 23 ‰ less depleted than previous studies (median = $-220$ ‰), but fall between $-310$ ‰ and $-196$ ‰ reported by Owen et al. (2016). In Fig. 4, the data from this study are compared with $\delta^{13}C_{CH4}$ and $\delta D_{CH4}$ values reported for $CH_4$ sourced from coal seams worldwide (Sherwood et al., 2017). The distribution of the data from this study sits within the

secondary microbial area of the CH₄ genetic diagram (see Fig. 4), which provides evidence that gas in the WCM has a secondary biogenic origin with a thermogenic component.

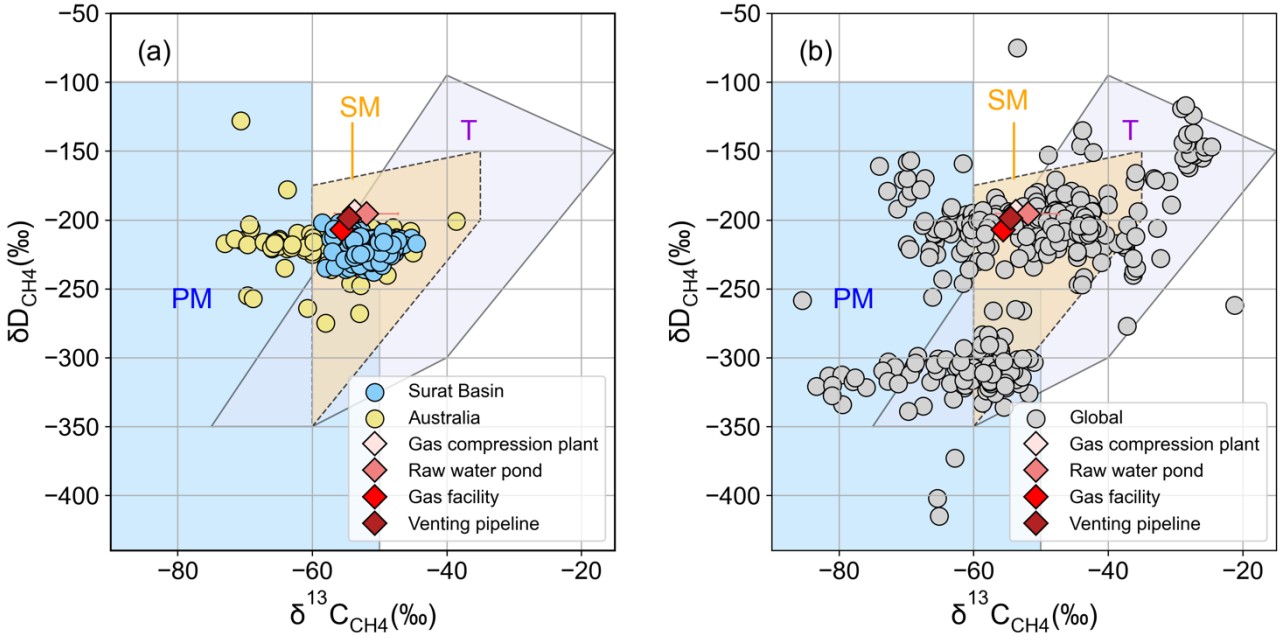

**Figure 4: A comparison of δ¹³C$_{CH4}$ and δD$_{CH4}$ of CSG from this study versus values from the Surat Basin, Australia wide and worldwide. Values for global measurements are shown in the inset CH₄ plot. All values are taken from Sherwood et al. (2017) and literature sources listed in Table 1. The gas genetic fields are taken from Milkov and Etiope (2018). PM: primary microbial; SM: secondary microbial; T: thermogenic.**

### 3.2.2 Coal mining

On 1 September 2019 samples were collected from a plume downwind of the Cameby Downs open-cut coal mine located approximately 16 km north-east of Miles (No. 6 in Table 2 and Figs. 1, 2 and 3). This is one of the largest coal mines in Australia with permission to extract up to 2.8 million tonnes per annum (Mtpa) of run-of-mine coal (Yancoal, 2018). The measured CH₄ mole fraction was between 2 ppm and 13 ppm north east of the coal mine. The sampled downwind plume from the Cameby Downs open-cut coal mine yielded δ¹³C$_{CH4}$ and δD$_{CH4}$ signatures of −60.0 ± 0.6 ‰ and −209.7 ± 1.8 ‰, respectively (see Table 2). These values are close to the values measured as part of this study from the ground seeps (abandoned coal exploration wells) (see Fig. 3) and sit within the range of the global and Australian CSG sectors (see Fig. 4). These results are expected because the δ¹³C$_{CH4}$ signatures from coal mines depend on coal rank and the process of secondary biogenic CH₄ generation (Zazzeri et al., 2016). Coals from the Cameby Downs mine are sub-bituminous to high-volatile bituminous (Hamilton et al., 2014) extracted from the relatively shallow Juandah measure (<200 m) in the Walloon Subgroup. Our results

are consistent with the values from Owen et al. (2016), which suggests the shallow coal measures have $\delta^{13}C_{CH4}$ and $\delta D_{CH4}$ signatures ranging from −80 ‰ to −50 ‰, and −310 ‰ to −210 ‰, respectively.

### 3.2.3 Ground and river seeps

Within the Surat Basin the origin of the CH4 associated with seeps mapped at various roadside locations or along the
Condamine River west of Chinchilla is poorly characterised (Day et al., 2013, 2015; Iverach et al., 2015; Nisbet et al., 2020). In our study during the 2018 and 2019 campaigns, two ground seeps and one river seep of CH4 were characterised (No. 7–9 in Table 2, Figs. 1, 2 and 3). Both ground seeps (believed to be coal exploration wells) are located along Green Swamp Road. At each site we sampled from near the plume centre (likely over the old borehole) to approximately 50 m away downwind to obtain a spread of CH4 mole fraction and isotopic composition data for Keeling plot analysis. The peak CH4 mole fractions
measured in the bag samples from seep A and seep B were 6 ppm and 18 ppm. Seep A had $\delta^{13}C_{CH4}$ and $\delta D_{CH4}$ signatures of −59.9 ± 0.3 ‰ and −185.0 ± 3.1 ‰. Seep B had $\delta^{13}C_{CH4}$ and $\delta D_{CH4}$ signatures of −60.5 ± 0.2 ‰ and −190.2 ± 1.4 ‰. The two ground seeps were also investigated in previous studies made by UNSW Sydney and RHUL, which reported $\delta^{13}C_{CH4}$ of −56.9 ‰ for gas collected from seep B (Day et al., 2015) and $\delta^{13}C_{CH4}$ of −60 ‰ (Iverach et al., 2014). The isotopic signatures indicate that the gas could originate from coal seams. We were able to visually confirm pieces of historical coal exploration and it was
stated in Day et al., (2015) that exploration drilling occurred at seep B during the 1970s. This is supported by the data available from the Queensland government, which shows a plugged and abandoned borehole at the same location. These likely coal seam sourced ground seeps have $\delta^{13}C_{CH4}$ and $\delta D_{CH4}$ signatures that align with the more depleted biogenic values (less than 55 ‰) of global coal gas and have slightly enriched $\delta D_{CH4}$ compared to Australian coal gas (see Fig. 4).

Many CH4 seeps have been located in the Condamine River, suggesting that the emitted CH4 is associated with coal seams in the area (Day et al., 2013; Department of Natural Resources and Mines, 2012). On 2 September 2019, we intersected CH4 plumes near the Chinchilla weir and measured CH4 mole fractions as high as 18 ppm in calm to light wind conditions (0–14 km h$^{-1}$). Gas samples had $\delta^{13}C_{CH4}$ and $\delta D_{CH4}$ signatures of −61.2 ± 1.4 ‰ and −225.1 ± 2.9 ‰, respectively. These values are similar to the results from the coal mine sampled in the study area (see Fig. 3). The $\delta^{13}C_{CH4}$ value is also consistent with the
results previously reported from gas samples collected in the Condamine River with values ranging from −63.4 ‰ to −59.3 ‰ (Department of Natural Resources and Mines, 2012). Iverach et al. (2017) proposed a hydrogeological conceptual model and CH4 production evolution model between the WCM and the overlying Condamine River alluvial aquifer indicating the upward migration of CH4 from the WCM. The relatively depleted $\delta^{13}C_{CH4}$ signature we measured is comparable to the values (−69.1 ‰) of CH4 believed to originate from shallow WCM in Iverach et al. (2017). The $\delta^{13}C_{CH4}$ and $\delta D_{CH4}$ signatures also align with
the values from Owen et al. (2016) showing CH4 from shallow coal measures (<200 m) have $\delta^{13}C_{CH4}$ and $\delta D_{CH4}$ signatures ranging from −80 ‰ to −50 ‰, and −310 ‰ to −210 ‰, respectively.

### 3.2.4 Abattoirs

High $CH_4$ mole fractions have been observed from intensive meat processing facilities in the study area (Nisbet et al., 2020). We sampled the plumes downwind of Beef City abattoir (Abattoir A) in 2018 and Oakey Beef Exports (Abattoir B) in 2019 (No. 10–11 in Table 2 and Figs. 1, 2, and 3).

The highest $CH_4$ mole fraction measured for the Beef City was 8.6 ppm, recorded on Toowoomba Cecil Plains Road 1.3 km downwind of the complex. The Beef City plume samples yielded a $\delta^{13}C_{CH4}$ signature of −46.0 ± 0.4 ‰. Beef City is an integrated feedlot and processing plant.

As part of the 2019 campaign, we sampled a $CH_4$ plume 1 km downwind of Oakey Beef Exports (Abattoir B). This plume extended northwest of the facility. The highest $CH_4$ mole fraction measured was 69.7 ppm, and the $\delta^{13}C_{CH4}$ signature was determined to be −44.5 ± 0.2 ‰. Emissions from Oakey Beef Exports have 4 potential sources, including a) the cattle themselves, b) emissions from anaerobic lagoons, c) emissions from biogas storage and combustion (from the facility exhaust stack), and d) by-products and animal wastes (paunch and manure). During the sampling night, smoke was observed continuously emitting from the stack associated with the main processing plant. We sampled in the centre line of that plume, but the other three potential sources must be considered, and it is likely that we sampled a mixed source plume. The processing plant is equipped with a waste-to-energy system that integrates biowaste treatment with biogas storage, processing and combustion. In the system, the biowaste is put in covered lagoons where anaerobic digestion occurs. In the anaerobic lagoons, concentrated anaerobic bacteria digest organic matter from Oakey Beef Export's biowaste to produce $CH_4$. During this biogas producing process, factors such as type of substrate, bacteria being used, and temperature can affect the isotopic signatures of produced gas. The generated biogas is stored in an onsite biogas storage tank and used to fuel the facility's boilers. The $\delta^{13}C_{CH4}$ signature of −44.5 ± 0.2 ‰ from this study is more enriched compared to the values from biogas plants in Heidelberg, Germany, which are fed by maize silage (−61.5 ± 0.1 ‰) and food waste (−64.1 ± 0.3 ‰) (Hoheisel et al., 2019) but close to maize-fed biogas plants in the UK (−45 ‰) (Bakkaloglu et al., 2020).

Values of $\delta^{13}C_{CH4}$ from both abattoirs are similar to values from global and Australian fossil fuels (Sherwood et al., 2017). In particular, the relatively enriched $\delta^{13}C_{CH4}$ compared to biogenic values suggests $CH_4$ could be derived from the incomplete combustion of biogas, which is similar with what has been reported (−48.1 ± 1.5 ‰) from measurement of a biogas power station in London, UK (Zazzeri, 2016). However, the $\delta D_{CH4}$ signature of −314.6 ± 1.8 ‰ from Oakey Beef Exports indicates a biological origin. These results are comparable with that of a piggery sampled in our study (see Fig. 3), the anaerobic digester values (−326.2 ‰) reported in NSW, Australia (Day et al., 2015) and closely resemble the values from a biogas generator ($\delta^{13}C_{CH4} = -51.8 ± 2.4$ ‰, $\delta D_{CH4} = -305.0 ± 12.0$ ‰) in Germany (Levin et al., 1993). On-site sampling at Oakey Beef Exports would be required to identify the exact source of the detected $CH_4$ plume. These abattoir readings highlight the problem of

using just $\delta^{13}C_{CH4}$ to attribute source. Using both $\delta^{13}C_{CH4}$ and $\delta D_{CH4}$ provides a more powerful discrimination between facility emissions from abattoirs and emissions from other gas sources.

### 3.2.5 Feedlot and grazing cattle

In the study area, we investigate the $\delta^{13}C_{CH4}$ and $\delta D_{CH4}$ signatures of $CH_4$ emitted from Stanbroke feedlot (No. 12 in Table 2, and Figs. 1 and 2) in 2018. The $\delta^{13}C_{CH4}$ and $\delta D_{CH4}$ signatures determined from Keeling plot had values of $-62.9 \pm 1.3$ ‰ and $-310.5 \pm 4.6$ ‰. The peak $CH_4$ mole fraction recorded was 3.2 ppm. In 2019 we sampled the $CH_4$ plume emitted from over 200 cattle grazing along the roadside between Dalby and Ranges Bridge (No. 13 in Table 2 and Fig. 3). The cattle were spread from immediately adjacent to the roadside to over 100 m away. The maximum $CH_4$ mole fraction value recorded for the grazing cattle plume was 7.4 ppm, and the $\delta^{13}C_{CH4}$ and $\delta D_{CH4}$ isotopic signatures were $-59.7 \pm 1.0$ ‰ and $-290.5 \pm 3.1$ ‰, respectively.

The isotopic signature of the cattle-produced $CH_4$ varies depending on the diet (Levin et al., 1993). In Queensland the typical cattle diet is predominantly C4 plant with forage, grain and supplements (McGinn et al., 2008). Specifically, due to differences in diet, the $\delta^{13}C_{CH4}$ and $\delta D_{CH4}$ signatures of cattle in the Surat Basin are in-between the values from Levin et al. (1993) ($\delta^{13}C_{CH4}$ = $-55.6 \pm 1.4$ ‰, $\delta D_{CH4}$ = $-295.0 \pm 10.0$ ‰, 60 – 80 % C4 diet) and Bilek et al. (2001) ($\delta^{13}C_{CH4}$ = $-70.6 \pm 4.9$ ‰, $\delta D_{CH4}$ = $-358.0 \pm 15.0$ ‰, 90 % C3 diet) (see Fig. 5). Compared to studies in the US, $\delta^{13}C_{CH4}$ signatures in our study are more depleted than those from cattle in Townsend–Small et al. (2015) ($\delta^{13}C_{CH4}$ = $-56.3$ ‰, $\delta D_{CH4}$ = $-283.0$ ‰, unspecified diet) and Townsend–Small et al. (2016) ($\delta^{13}C_{CH4}$ = $-56.2$ ‰, $\delta D_{CH4}$ = $-302.0$ ‰, unspecified diet) (see Fig. 5). Both the feedlot and grazing cattle signatures determined as part of this study are generally consistent with values for ruminants around the globe and in other areas of Australia (see Table 1).

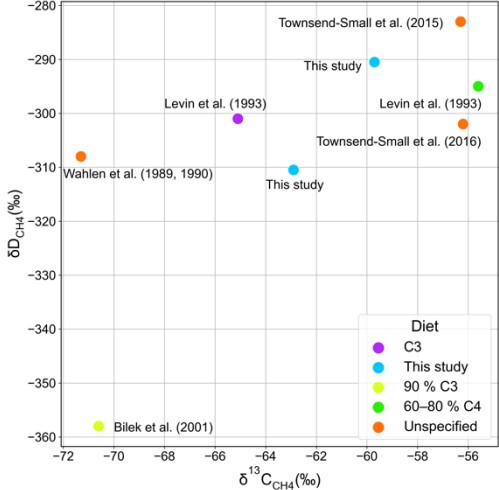

**Figure 5: A dual isotope plot comparing the $\delta^{13}C_{CH4}$ and $\delta D_{CH4}$ for cattle from this study with the values reported in the literature (indicated next to the data points).**

**3.2.6 Piggery**

A $CH_4$ plume was sampled 600 m downwind of Albar Piggery in 2019. This plume had a distinctive smell and a warmer temperature compared to the surrounding ambient air, indicating that the piggery was heated. The maximum $CH_4$ mole fraction measured was 14.7 ppm, and the $\delta^{13}C_{CH4}$ and $\delta D_{CH4}$ signatures were $-47.6 \pm 0.2$ ‰ and $-300.1 \pm 2.6$ ‰, respectively (No. 14 in Table 2, Figs. 1, 2 and 3). These $\delta^{13}C_{CH4}$ and $\delta D_{CH4}$ signatures are close to those reported by Levin et al. (1993) in Germany for lower pile of manure ($\delta^{13}C_{CH4} = -45.5 \pm 1.3$ ‰ and $\delta D_{CH4} = -297.0 \pm 6.0$ ‰). The $\delta^{13}C_{CH4}$ and $\delta D_{CH4}$ values also closely resemble our results from the abattoirs (Fig. 3).

**3.2.7 Landfill**

Gas samples collected downwind of the Chinchilla landfill had a $CH_4$ mole fraction range from 1.8 to 2.1 ppm, and a Keeling plot best fit $\delta^{13}C_{CH4}$ value of $-52.1 \pm 3.6$ ‰ (No. 15 in Table 2 and Figs. 1, 2 and 3). In general, the determined $\delta^{13}C_{CH4}$ value in this study falls into the range of international and Australian $CH_4$ sourced from waste (Table 1, Sherwood et al., 2017). The isotope ratio of $CH_4$ in this landfill is less depleted than the mean values reported ($-56.5$ ‰ for surface and $-58.7$ ‰ for waste) of the active landfill in Ipswich, Queensland (Obersky et al., 2018) and those reported from Europe (Hoheisel et al., 2019; Xueref−Remy et al., 2020; Zazzeri et al., 2015) possibly due to $CH_4$ oxidation by aerobic bacteria in cover soils. Similarly, relatively enriched $\delta^{13}C_{CH4}$ values were also identified from older, closed landfills in the UK (Bakkaloglu et al., 2020; Lowry et al., 2020). Our result also closely resembles the value measured by Day et al. (2015), who reported $-53.0$ ‰ for a landfill in New South Wales, Australia and results from the upper layers of waste ($-52.0$ ‰) in Germany (Levin et al., 1993).

**3.2.8 Wastewater treatment plant (WWTP)**

On 2 September 2019 we sampled a plume immediately adjacent to the Miles wastewater treatment plant along Waterworks Road. This plume had a maximum $CH_4$ mole fraction reading of 19.6 ppm, and $\delta^{13}C_{CH4}$ and $\delta D_{CH4}$ signatures of $-47.6 \pm 0.2$ ‰ and $-177.3 \pm 2.3$ ‰ (No. 16 in Table 2 and Figs. 1, 2 and 3), respectively. In Australia the $\delta^{13}C_{CH4}$ of $CH_4$ emissions from the waste sector ranges from $-58.8$ ‰ to $-44.0$ ‰ with a median of $-50.4$ ‰ (AGL Energy Limited, 2015; Day et al., 2015; Obersky et al., 2018; Sherwood et al., 2017), the $\delta^{13}C_{CH4}$ $-47.6 \pm 0.2$ ‰ determined for the Miles wastewater treatment plant is consistent with past results. However, the $\delta^{13}C_{CH4}$ signature is less depleted than the wastewater treatment plant values of $-51.3 \pm 0.2$ ‰ measured in Heidelberg, Germany (Hoheisel et al., 2019), $-52.3$ ‰ in Cincinnati, USA (Fries et al., 2018) and $-59.2$ ‰ to $-50.7$ ‰ in London, UK (Zazzeri, 2016) for anaerobic treatment systems. The result is similar to the measurements made by Townsend−Small et al. (2012) from two wastewater treatment plants ($-46.3$ ‰ and $-47$ ‰) in the metropolitan area of Los Angeles, USA and result from aerobic digestion tank of WWTP ($-45.5$ ‰) in Tokyo, Japan (Toyoda et al., 2011). Both Townsend−Small et al. (2012) and Fries et al. (2018) found a more depleted $\delta D_{CH4}$ for wastewater treatment plants in Los Angeles ($-298$ ‰) and Cincinnati ($-325$ ‰) compared to our result. Toyoda et al. (2019) suggested that the relatively enriched $\delta^{13}C_{CH4}$ signature could be due to aerobic digestion. A better understanding of the $CH_4$ from wastewater treatment plants,

especially for different treatment processes (anaerobic or aerobic), in Australia is needed as it is proven to be a non-negligible source of $CH_4$ emission in urban areas.

### 3.3 Discriminating between isotopic signatures from various sources: uniqueness and overlaps.

Various studies have pointed out that there are large overlaps in $CH_4$ isotopic signatures, compromising the use of isotopic constraints in models estimating $CH_4$ emissions (Feinberg et al., 2018; Milkov and Etiope, 2018; Sherwood et al., 2016, 2017). Figure 6 displays probability distributions of $\delta^{13}C_{CH4}$ and $\delta D_{CH4}$ for fossil fuel and modern microbial processes (with their respective subcategories) in Australia (Table 1 and Sherwood et al., 2017) and around the globe (Sherwood et al., 2017). Global coal gas $\delta^{13}C_{CH4}$ has a bimodal distribution and a relatively wide range spanning from −85.5 ‰ to −16.8 ‰. In Australia, coal gas has a unimodal distribution of $\delta^{13}C_{CH4}$ ranging from −76.8 ‰ to −30.3 ‰ with a more depleted median of −54.3 ‰ due to high amount of microbial gases. Almost half of the widely spread values of coal gas have a range that overlaps with the distributions of other microbial processes. Specifically, global $\delta^{13}C_{CH4}$ of cattle varies from −71.3 ‰ to −50.3 ‰ with a median of −66.5 ‰; values for Australia range from −70.6 ‰ to −49.0 ‰ with a median of −61.5 ‰. The more enriched isotopic values found in Australian cattle are likely due to higher proportions of a C4 diet (Levin et al., 1993; McGinn et al., 2008) in these tropical herds, raised on C4 grasslands and with maize supplements.

In this study, $\delta^{13}C_{CH4}$ signatures determined from CSG processing and production infrastructures and seeps varied from −61.2 ‰ to −50.9 ‰ with a median of −55.6 ‰. This range is far narrower than the global distribution of $\delta^{13}C_{CH4}$ from coal presented in Sherwood et al. (2017) (Fig. 6), or those determined from gas and water well measurements (Baublys et al., 2015; Hamilton et al., 2014, 2015). The median of CSG $\delta^{13}C_{CH4}$ signature is about 6 ‰ more enriched than the $\delta^{13}C_{CH4}$ signature of the cattle (which ranges from −62.9 ‰ to −59.7 ‰) and about 6 ‰ more depleted than that of waste (which ranges from −52.1 ‰ to −47.6 ‰). These similar or overlapping $\delta^{13}C_{CH4}$ values for different sources mean that in areas with multiple sources like the Surat Basin CSG fields, we cannot assign a source to a plume using $\delta^{13}C_{CH4}$ alone.

Previously, Maher et al. (2014) undertook a mobile $CH_4$ survey using a Picarro G2201-i CRDS in the Tara region of the Surat Basin. Based on isotopic measurements, they divided the region into a CSG field sub-region (−54.7 ‰) and a non-CSG field sub-region (−47.4 ‰). These results were blended signatures produced by combining all data within each sub-region. As the individual plume analyses shown in Table 2 and Fig 3 demonstrate, single sub-region values cannot be used to isolate CSG emissions from mixtures of other sources, as many sources (CSG, seeps, agricultural) with similar $\delta^{13}C_{CH4}$ signatures co-exist in the CSG sub-region. As shown in this study, attributing $CH_4$ emissions to CSG sources in the area requires careful analysis using a combination of insights.

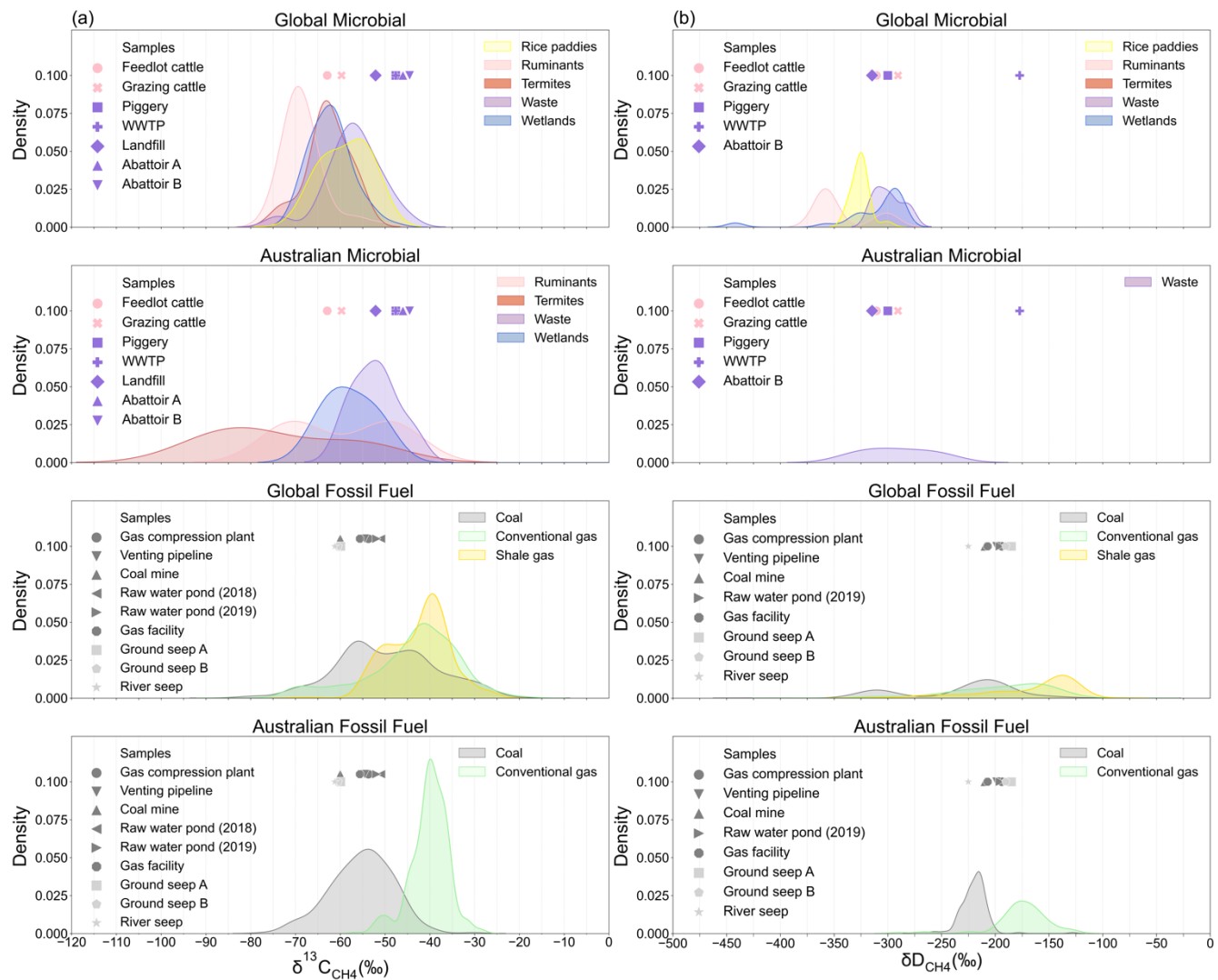

**Figure 6: Probability density plot of literature values (globally and from Australia) for (a) $\delta^{13}C_{CH4}$ and (b) $\delta D_{CH4}$ and results from this study (global values taken from Sherwood et al. (2017) and literature sources listed in Table 1).**

Hatch et al. (2018) have also studied $CH_4$ emissions in the Surat Basin CSG field using a Picarro G2201-i CRDS. The objective of their study was to distinguish between CSG $CH_4$ (thought initially to be thermogenic origin) and biogenically sourced $CH_4$. They suggested that $\delta^{13}C_{CH4}$ surveys would not be effective in the Surat Basin, due to small differences of isotopic signatures between the sources of interest. However, our findings are less pessimistic about the usability of $\delta^{13}C_{CH4}$. In the right settings, $\delta^{13}C_{CH4}$ can be used as part of two endmember mixing studies, especially when there are extreme endmembers in the mixed air sample. This is highlighted for the two abattoirs. If the $CH_4$ emissions downwind of the abattoirs were due to enteric fermentation a $\delta^{13}C_{CH4}$ signature of −63.0 ‰ to −60.0 ‰ would have been recorded. However, at both abattoirs the plumes

had isotopic signatures of −46.0 ‰ to −44.5 ‰ (No. 10–11 in Table 2, and Figs. 1, 2 and 3), so clearly the bulk of the plume being emitted from these facilities is not due to direct cattle emissions and is suspected to be related to the processing of waste meat products, animal wastes, or a mixture of enteric fermentation and biogas combustion. These results highlight the need for further studies of emissions from large feedlots and abattoirs.

This study shows that the combined use of $\delta^{13}C_{CH4}$ and $\delta D_{CH4}$ provides critical insights into determining the sources of the mapped plumes. In Fig. 3, it is clear that sources such as CSG processing, seeps, ruminants and waste are in distinct dual isotope clusters. In the study area, livestock has relatively depleted $\delta^{13}C_{CH4}$ signatures that are close to CSG sources. However, the $\delta D_{CH4}$ signatures from cattle, the piggery and the abattoir are 100 ‰ more depleted than the other sources, which

successfully sets them apart from CSG sources. We expect the use of $\delta^{13}C_{CH4}$ and $\delta D_{CH4}$ to reduce uncertainties in interpreting air samples from mixed sources. These results will facilitate improved interpretation of airborne measurements where elevated $CH_4$ mole fraction readings are due to two or more sources of $CH_4$.

Establishing the source signatures for the 16 sources in this study required many weeks in the field and the laboratory. Ensuring

statistically robust source signature population statistics in a timely manner requires the development of infield methods. Recent advances in the application of moving Keeling and Miller–Tans methods (Assan et al., 2018; Menoud et al., 2020; Röckmann et al., 2016; Vardag et al., 2016) used in conjunction with portable laser adsorption spectroscopy systems has the potential to provide better source signature population statistics for $\delta^{13}C_{CH4}$ (Kelly and Fisher, 2018; Lu et al., 2019). However, equipment advances are required before we can do in field $\delta D_{CH4}$ measurements, and as this study has demonstrated both

$\delta^{13}C_{CH4}$ and $\delta D_{CH4}$ are needed for improved source identification. These results also demonstrate the value of collating global databases (Sherwood et al., 2017).

## 4 Summary

In 2018 and 2019, a mobile system was used to map the $CH_4$ mole fractions and identify various $CH_4$ sources in the south-eastern Surat Basin CSG fields in Queensland, Australia. Generally, the $\delta^{13}C_{CH4}$ and $\delta D_{CH4}$ signatures determined from isolated

plumes mapped during our 2018 and 2019 campaigns agree with values reported in the literature (Table 1 and Fig. 6). We present the $\delta^{13}C_{CH4}$ isotopic signatures for 16 plumes and the $\delta D_{CH4}$ isotopic signatures for 13 plumes, from the analyses of over 160 air samples. Despite the size of the data set, for many sources only a single isotopic signature has been determined. However, this single isotopic value represents the first recorded isotopic signature for some sources (e.g., abattoirs and piggeries) in Australia. Generally, the $\delta^{13}C_{CH4}$ and $\delta D_{CH4}$ signatures determined from isolated plumes mapped during our 2018

and 2019 campaigns agree with values reported in the literature (Table 1 and Fig. 6). More investigations in Australia are needed for further characterisation of other sources, both those listed in the UNFCCC inventory classifications and natural. There is also a need for further studies to characterise the temporal and spatial variability of all sources, climatic and seasonal

influences, and procedural repeatability. Ideally, further sampling should be undertaken in collaboration with the operators of each facility, so that samples can be collected closer to the source, removing all uncertainty in the origin of the CH4. This study has made a contribution to the $\delta^{13}C_{CH4}$ and $\delta D_{CH4}$ signatures from different sources in Australia and internationally. We also show that the $\delta^{13}C_{CH4}$ and $\delta D_{CH4}$ signatures of atmospheric CH4 can provide crucial information for characterising closely located sources. Combined $\delta^{13}C_{CH4}$ and $\delta D_{CH4}$ signatures separate cattle (both feedlot and pasture) from natural gas seeps and all produced gas sources when measured as unmixed plumes. The dual isotopes $\delta^{13}C_{CH4}$ and $\delta D_{CH4}$ also separate natural gas seeps, or emissions from the nearer surface portion of the WCM from the production interval within the same coal measure. Results from the piggery and abattoirs cluster together, and these two sources have a $\delta^{13}C_{CH4}$ and $\delta D_{CH4}$ signature set that is distinct from all other sources sampled.

Previous studies have indicated that using a single tracer (e.g., $\delta^{13}C$) is effective only for single CH4 emission sources, where a single source is mixed with background air. Challenges emerge when several sources exist in the same region (Hatch et al., 2018; Mielke–Maday et al., 2019; Townsend–Small et al., 2015). Within the Surat Basin the range of $\delta^{13}C_{CH4}$ extends from −63 ‰ to −45 ‰. When considering only $\delta^{13}C_{CH4}$, plumes from abattoirs, piggeries, wastewater treatment plants and conventional gas pipelines cannot be differentiated from each other. The $\delta^{13}C_{CH4}$ signatures from CSG sources overlap with signatures expected from landfills. Source attribution using $\delta^{13}C_{CH4}$ signatures alone must be done with local context insights. Without knowing distance to a source or sources, wind speed and direction information, temperature, and mixing layer details, it is not possible from $\delta^{13}C_{CH4}$ signatures alone to separate cattle (both feedlot and pasture) emissions from shallow open-cut coal mines, natural seeps from the upper portion of the Walloon Coal Measures, or many other natural biological sources. However, the distinction of CSG CH4 emissions is possible using $\delta D_{CH4}$, because when it is combined with the $\delta^{13}C_{CH4}$ signature it plots in an isolated cluster in Fig. 3.

It is clear that the separation in the dual isotope plot prompts an in-depth investigation of the feasibility for constraining local and regional-scale emissions. Time series measurements of both $\delta^{13}C_{CH4}$ and $\delta D_{CH4}$ signatures should also provide further insights for the ongoing rise of the CH4 mole fraction both regionally and globally.

## Appendix A

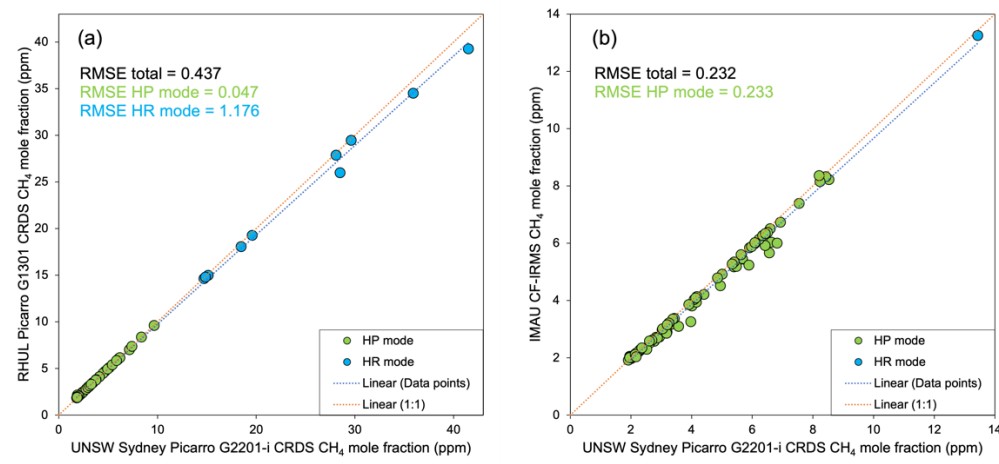

**Figure A1: Cross-plots of CH$_4$ mole fraction values measured from bag samples using the UNSW Sydney Picarro G2201-i CRDS and the RHUL Picarro G1301 CRDS (a) in 2018 and the UNSW Sydney Picarro G2201-i CRDS and IMAU CF-IRMS (b) in 2019.**

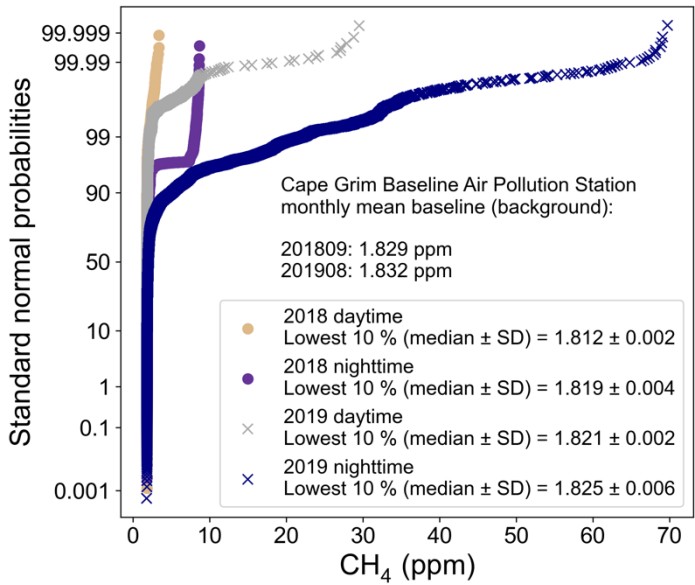

**Figure A2: Probability of CH$_4$ mole fraction values measured from daytime and nighttime surveys in 2018 and 2019. The median of lowest 10 % ± standard deviation (SD) was calculated to represent the background ambient air.**

Each plume set of air samples (blue dots) was analysed using the Keeling plot method. The results are shown in Figs. A3–A7. The blue lines are the Bayesian linear regression posterior mean fits, and the 95 % Bayesian credible intervals are shown in purple.

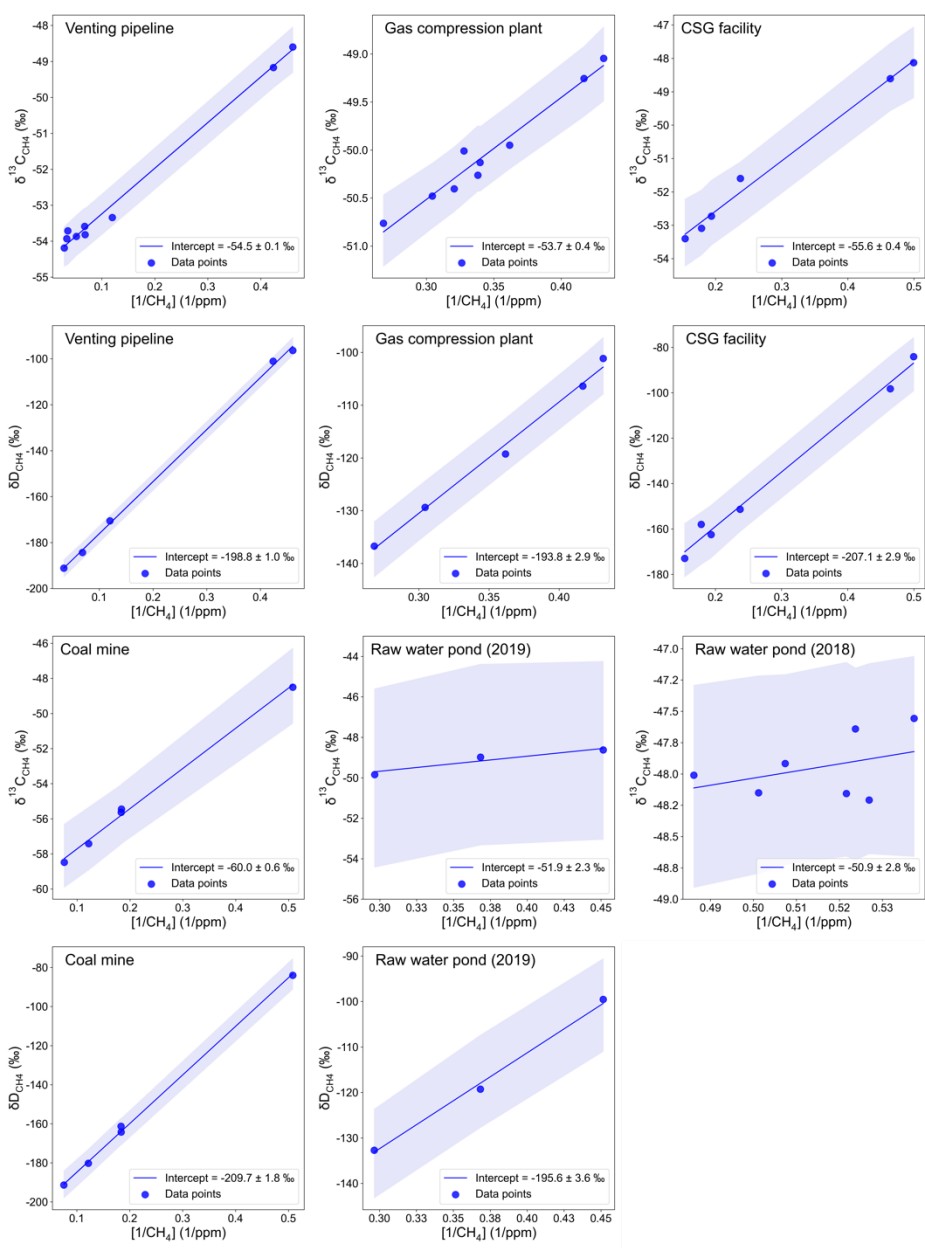

**Figure A3: Keeling plots of all data from CSG infrastructures and a coal mine analysed using Bayesian linear regression. Upper panels show the results for δ¹³C and lower panels show the results for δD.**

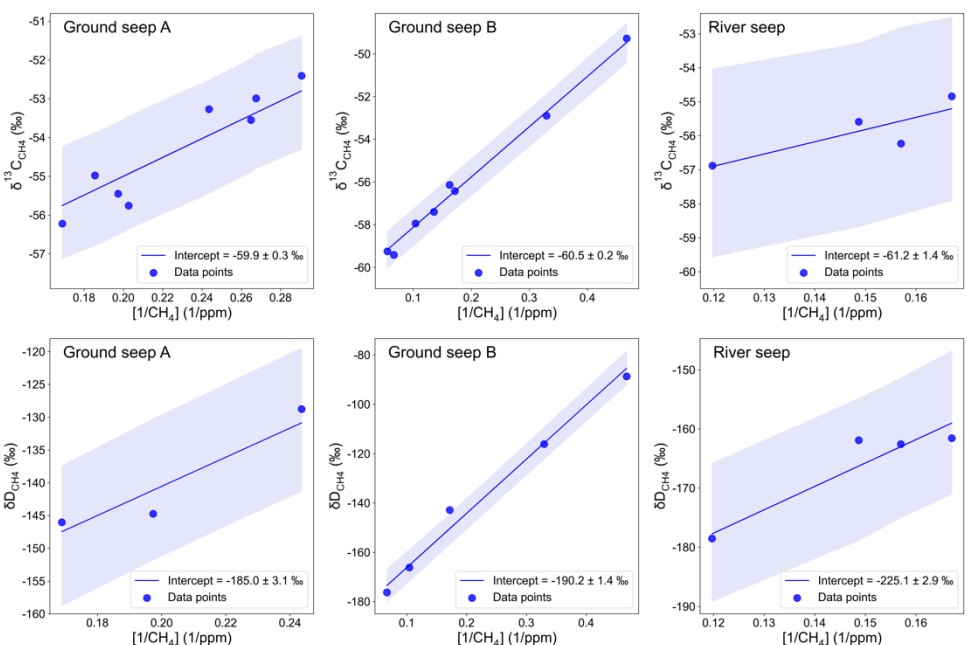

**Figure A4: Keeling plots of all data from ground and river seeps analysed using Bayesian linear regression. Upper panels show the results for δ¹³C and lower panels show the results for δD.**

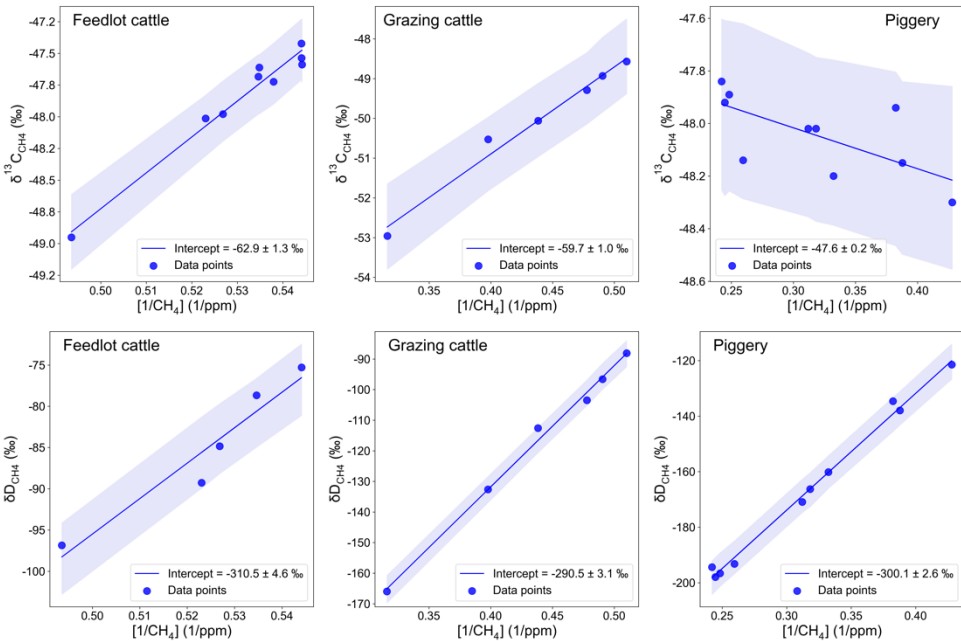

**Figure A5: Keeling plots of all data from agricultural sources analysed using Bayesian linear regression. Upper panels show the results for δ¹³C and lower panels show the results for δD.**

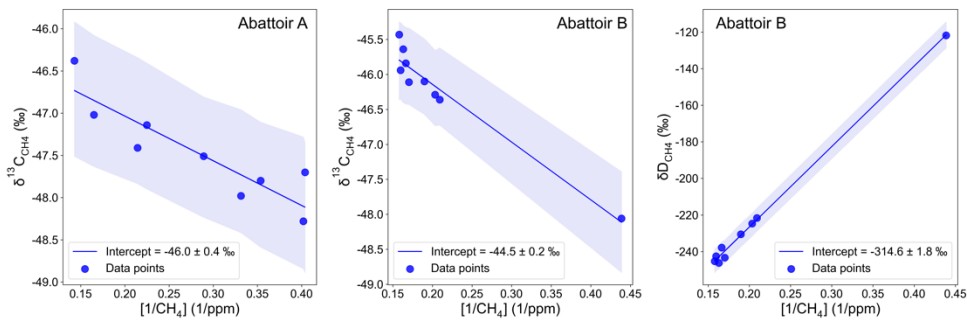

**Figure A6: Keeling plots of all data from export abattoirs analysed using Bayesian linear regression. Left ($\delta^{13}$C) panel shows the result for abattoir A and the middle ($\delta^{13}$C) and right panel ($\delta$D) show the results for abattoir B.**

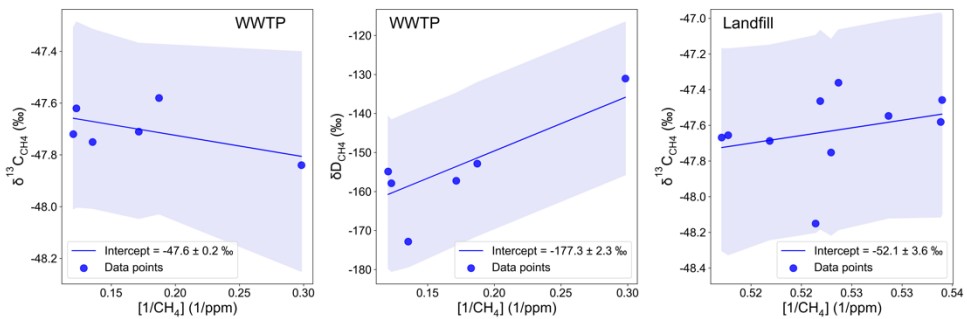

**Figure A7: Keeling plots of all data from the WWTP and landfill analysed using Bayesian linear regression. Left ($\delta^{13}$C) and middle ($\delta$D) panels show the results for WWTP and right panel ($\delta^{13}$C) shows the result for landfill.**

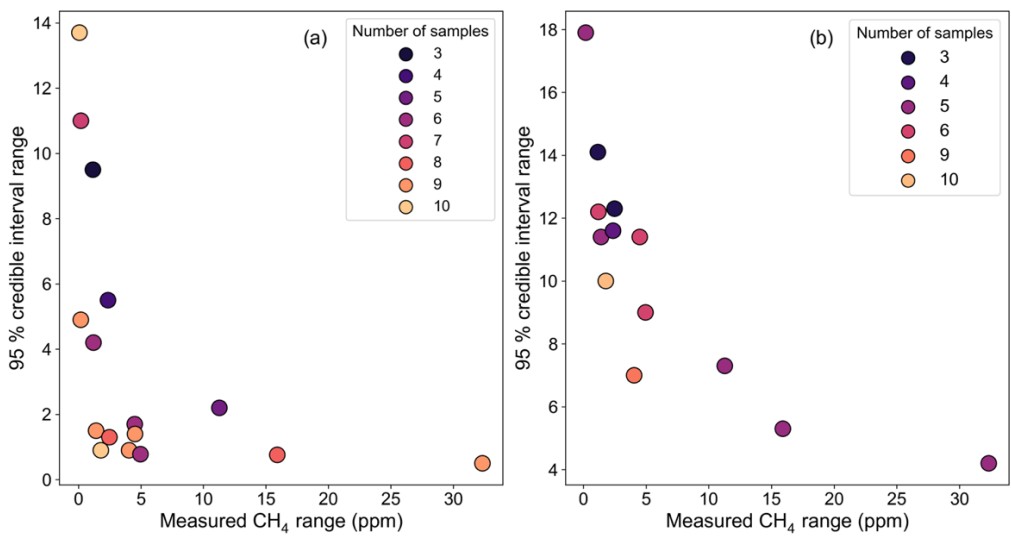

**Figure A8: Dependency between the 95 % credible interval range of $\delta^{13}$C (a) and $\delta$D (b) derived from the Keeling plot method and number of samples and measured CH$_4$ mole fraction range from the corresponding measured sources.**

**Data availability.**

All data are available from the corresponding author upon request.

**Author contributions.**

Xinyi Lu: Investigation, Formal analysis, Visualisation, Validation, Writing – original draft, Writing – review & editing.
Stephen J. Harris: Investigation, Formal analysis, Writing – review & editing. Rebecca E. Fisher: Supervision,
Conceptualisation, Methodology, Formal analysis, Writing – review & editing. James L. France: Conceptualisation, Writing
– review & editing. Euan G. Nisbet: Conceptualisation, Writing – review & editing. David Lowry: Conceptualisation, Writing
– review & editing. Thomas Röckmann: Formal analysis, Writing – review & editing. Carina van der Veen: Formal analysis.
Malika Menoud: Formal analysis, Writing – review & editing. Stefan Schwietzke: Conceptualisation, Methodology,
Investigation, Validation, Writing – review & editing. Bryce F. J. Kelly: Conceptualisation, Funding acquisition, Supervision,
Project administration, Methodology, Validation, Investigation, Formal analysis, Writing – review & editing.

**Competing interests.**

The authors declare that they have no conflict of interest.

**Acknowledgements.**

This work was under the support of the Climate and Clean Air Coalition (CCAC) Oil and Gas Methane Science Studies (MSS),
hosted by the United Nations Environment Programme. Funding was provided by the Environmental Defense Fund, Oil and
Gas Climate Initiative, European Commission and CCAC. The authors thank the MSS Science Advisory Committee, the MSS
Technical Working Group as well as Christopher Konek for valuable suggestions and comments on this project. We also thank
Bruno Neininger and Jorg Hacker for useful inputs during the field campaign and Lisa Williams for reviewing and editing the
final paper. The authors appreciate the insightful feedbacks from the reviewers which helped improving the overall quality of
the paper.

**Financial support.**

The financial support for this study has been provided by Environmental Defense Fund, Oil and Gas Climate Initiative,
European Commission and CCAC (United Nations Environment Programme grants DTIE18-EN067 and DTIE19-EN0XX;
UNSW Sydney grant numbers RG181430 and RG192900). Xinyi Lu is supported in part by UNSW–China Scholarship
Council (CSC) Scholarship. Malika Menoud was supported by the European Union's Horizon 2020 research and innovation

programme under the Marie Sklodowska-Curie grant agreement No 722479. Stefan Schwietzke was funded by the Robertson Foundation.

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
