# Peer review of "Isotopic Signatures of Major Methane Sources in the Coal Seam Gas Fields and Adjacent Agricultural Districts, Queensland, Australia"

_Atmospheric Chemistry and Physics, 2021_

## Referee Comment (RC1)

Review of the manuscript "**Isotopic Signatures of Major Methane Sources in the Coal Seam Gas Fields and Adjacent Agricultural Districts, Queensland, Australia**" by Xinyu Lu et al.

**Short summary of the manuscript**

This paper presents results from campaigns of in-situ $CH_4$ mole fraction measurements and of air samplings for subsequent isotope analyses. The authors characterized carbon and hydrogen stable isotope signatures of various $CH_4$ sources in the Surat Basin, Australia and compared the values with those in previously reported literature.

**General comment**

This is a good piece of work showing 2-year campaign measurement results. Previously, Sherwood et al. (2017) compiled thousands of isotope signature data of various $CH_4$ sources, but availability of such data is still limited for some regions or source types. This study will be acknowledged for complement of the available dataset and for important information for isotope-based top-down estimate of the Australian $CH_4$ emissions. This work is well within the scope of the journal and has significance in the study field. For publication of this manuscript, I would like to encourage the authors major revisions to consider my comments below.

**1. Representativeness of the isotope signatures presented in this study**

This study presents isotope signatures of various sources in the Surat Basin. The results showed that the estimated isotope signatures of same source type sometimes differ from site to site. In such field studies, critically important is how representative the result is. Previous studies have inferred that $\delta^{13}C$ and $\delta D$ signatures of $CH_4$ emitted from some sources vary considerably with time and space. I am very curious to discussion on how representative the isotope signatures obtained in this study are. Such discussion is crucial when the results are considered for constraints to the regional $CH_4$ budget. There are two points I can come up with. First, whether the set of air samples were collected from downwind of the major emission locations of the source. The higher fraction of emission at the location among the total emission is, the more representative the results would be. Second, how large variability among isotope signatures from a single source type is. For instance, the results indicate that isotope signatures of $CH_4$ of CSG origin vary by 5‰ in $\delta^{13}C$ and by 20‰ in $\delta D$, depending on sampling location/time. But the variability is much smaller for ground seeps in $\delta^{13}C$, which may suggest the source isotope signatures are relatively constant/uniform. Such discussion could be made not only from the results from the present study but also from those from the earlier studies (Table 1). I would like to see more enriched discussions about what we can learn about representativeness from this study and give some suggestions about better sampling strategy for a similar study in future. It could affect discussions in section 3.3, because distinctiveness of isotope signatures of different source types depends on this issue. For instance, some part of $\delta^{13}C$ and $\delta D$ signatures overlap between fossil fuel and biogenic sources according to the Sherwood et al. (2017) database, but their global representative values are considered to be sufficiently different so that we can examine partition estimates for contributions of both sources to atmospheric $CH_4$ variations.

**2. Data analysis**

The authors employ the Miller-Tans plot. I could not however understand advantage of the analysis over the traditional Keeling plot. Keeling plot assumes that the background atmosphere is constant over the time period of interest, and thus provides a much simpler framework and data interpretation. In contrast, as shown by Miller and Tans (2003), Miller-Tans plot is useful when one needs to assume that the background atmospheric condition varies with time at significant level of magnitude. To my understanding, Miller-Tans plot should be chosen only for special cases, where constant background cannot be assumed. From this point of view, the authors argument "The background air normally does not change during this period but using the Miller-Tans approach is a safeguard against any variability." (P11 L295) seems strange to me. Not only for the original Keeling (1958) paper (cited in the manuscript), the authors might revisit Miller and Tans (2003) for different ways of use of the two plots. Pataki et al. (2003) might also help for limitation of the Keeling plot.

In the case where the authors continue use of the Miller-Tans plot, they should clearly present how the time-varying background mole fractions and delta values are given for individual data points. It is not presented explicitly in the current manuscript. I suppose that it would for instance call for on-site continuous measurements in the upwind. Miller and Tans (2003) applied curve-fitting to their time-series $CO_2$ data, and similar examples for $CH_4$ were given by Umezawa et al. (2012).

Additional useful reference is Zobits et al. (2006), who investigated difference of source isotope signatures between Keeling and Miller-Tans plots and between ordinary square and geometric mean regressions. They found that, for same conditions, both plots return practically identical source isotope signatures. However, when the range of mole fraction is relatively small, geometric mean regression method could cause a bias in the estimated source signature. Since this study targets relatively wide range of mole fraction variation, I suppose such a bias is small, but the authors could examine with their data and/or justify validity of their regression approach. According to the text, the "Bayesian regression" used in this study seems to be similar to geometric mean regression that could cause a bias for small range of mole fraction. Please explain more to justify the analysis method.

In summary, there would be no difference in estimated source signatures from both plots if constant background is assumed. However, unless the reason why Miller-Tans plot is needed is clearly given, I see that the authors make the analytical methodology complicated without necessity. And it is not logically clear why they chose the regression method. I would like to suggest the authors to revisit previous studies carefully and rewrite the corresponding section.

Pataki, D. E. et al. (2003), The application and interpretation of Keeling plots in terrestrial carbon cycle research. Global Biogeochem. Cycles, 17(1), 1022, https://doi.org/10.1029/2001GB001850.

Umezawa, T. et al. (2012), Contributions of natural and anthropogenic sources to atmospheric methane variations over western Siberia estimated from its carbon and hydrogen isotopes, Global Biogeochem. Cycles, 26, GB4009, https://doi.org/10.1029/2011GB004232.

Zobits, J. M. et al. (2006), Sensitivity analysis and quantification of uncertainty for isotopic mixing relationships in carbon cycle research. Agricultural and Forest Meteorology, 136, 56–75, https://doi.org/10.1016/j.agrformet.2006.01.003.

**3. Description of the instrument performance**

The authors should present their performance tests of the instruments (Los Gatos and Picarro) made before the campaigns. It seems that the current descriptions rely on information provided by the product companies only, which is performance of their standard products at shipment from factory, but we know that there are differences among products and importantly that stability of those instruments (e.g. during shipment, repetitive power on/off, change in temperature/humidity) has not been well established. It is important to show readers sufficient key information on validity of their measurements, independent from that from the provider.

**Specific Comments**

P1 L14: "The use of…" This sentence needs rewriting. It is not clear how the authors consider difference of the three verbs. I think that (use of) $\delta^{13}C$ and $\delta D$ can "help us" identify a specific source "if potential sources are all characterized in $\delta^{13}C$ and $\delta D$ signatures."

P1 L16: It is not clear how different "discriminate" is from for instance "distinguish between" in the earlier sentence.

P2 L40: "e.g." should be added before "Nisbet et al. 2020", otherwise additional reference that cover "greenhouse gases" is needed. I think the reference is for $CH_4$ only.

P2 L60: The $CH_4$ increase over the industrial era and stagnation during the early 2000s are phenomena with totally different time scales.

P3 L65: There are indeed many references that addressed different time phases of the atmospheric $CH_4$ increase. First, I would like to suggest the authors to cite references that substantially contributed to the present study only. Did all these references contributed equally to the present study? Second, as written in the sentence, such debate has continued over the last decades, while the references are all relatively new. It might give wrong impression that the problem is new. Some "old" but key references, for instance Steele et al. (1992), Dlugokencky et al. (1998), Bousquet al. (2006), Simpson et al. (2012) and others, could be considered. Additionally, a recent study Chandra et al. (2021) present conclusion similar to that by Jackson et al. (2020). That said, choice of references is up to the authors.

Steele, L., Dlugokencky, E., Lang, P. et al. Slowing down of the global accumulation of atmospheric methane during the 1980s. Nature 358, 313–316 (1992). https://doi.org/10.1038/358313a0.

Dlugokencky, E., Masarie, K., Lang, P. et al. Continuing decline in the growth rate of the atmospheric methane burden. Nature 393, 447–450 (1998). https://doi.org/10.1038/30934.

Bousquet, P., Ciais, P., Miller, J. et al. Contribution of anthropogenic and natural sources to atmospheric methane variability. Nature 443, 439–443 (2006). https://doi.org/10.1038/nature05132.

Simpson, I., Sulbaek Andersen, M., Meinardi, S. et al. Long-term decline of global atmospheric ethane concentrations and implications for methane. Nature 488, 490–494 (2012). https://doi.org/10.1038/nature11342.

Chandra, N., P. K. Patra, J. S. H. Bisht, et al. Emissions from the Oil and Gas Sectors, Coal Mining and Ruminant Farming Drive Methane Growth over the Past Three Decades. Journal of the Meteorological Society of Japan (2021). https://doi.org/10.2151/jmsj.2021-015.

P3 L69: "natural fossil fuel source" looks strange. All fossil fuels are of natural origin in nature. The difference is just emission takes place by nature or by human. Etiope and colleagues have used the term "geological". Please rephrase.

P3 L69: What is the current global fraction of emissions from unconventional sources in the total fossil fuel related $CH_4$ emissions?

P4 L121: Same comment as that for P3 L65.

P8 L219: What is the actual precision evaluated by the study team and expected during the campaign? Please see my earlier comment.

P9 L225: The same question as that for P8 L219. The information here seems to be identical to those on the Data Sheet provided on the Picarro website (https://www.picarro.com/support/library/documents/g2201i_analyzer_datasheet). These are general instrument performance when it is shipped from factory. I suppose that the study team did not use the instrument as delivered but carried out series of evaluations before the campaign. The authors should present those results that better represent the actual performance during the campaign.

P9 L231: The more details of the time lag correction should be presented. How did you estimate the time lag? Is it time dependent or constant? How long is it on average?

P9 L243: Please describe the value explicitly. What is the exact value and uncertainty measured by another laboratory? I believe the measurement was made by INSTAAR, not NOAA/GML.

P9 L247: Was the driving speed adjusted specifically for the different instruments (e.g. flow rate)? If so, additional information will help readers.

P10 L267: Here the authors refer to the WMO X2004A scale traceable by the gas cylinders provided by NOAA. In the section before, they wrote "WMO scale" only. Reading these sentences, I cannot be sure that the mole fraction values from CSIRO and RHUL are on the identical scale. Please sort this out so that use of one single measurement scale is clear throughout the paper. Though the principle methodology is still valid, since the original WMO X2004 scale was updated to WMO X2004A in 2015, the reference Dlugokencky et al. (2005) no longer represents the current WMO scale exactly. I would suggest to additionally cite the NOAA website (https://www.esrl.noaa.gov/gmd/ccl/ch4_scale.html) or the latest WMO GAW Report No. 255 (https://library.wmo.int/index.php?lvl=notice_display&id=21758#.YE7BLS9h0UE). It should be somewhere noted that the current WMO scale cover up to ~5900 ppb and many measurements presented in this study are calculated by extrapolation of the scale. The abbreviation "NOAA" appeared earlier.

P10 L276: I would suggest a sentence like "For the subsequent IRMS measurement, $CH_4$ in sample air in most bags were preconcentrated for 10 minutes at…, but that in samples with $CH_4$ mole fraction larger than 6 ppm reported by RHUL were processed for shorter time in order to…" if I understand correctly. I suppose that "sampled" in the original sentence is not

just collecting air but preconcentration of $CH_4$ from sample air, as written in the following sentence.

P10 L283: What does "UNSW" stands for?

P11 Section 2.4: Please see my earlier comment.

P11 Section 3.1: More than half of sentences of this paragraph are for explaining the switch from daytime to nighttime sampling. I do not think the title of the section represent the content well. A problem of nighttime samplings might be darkness, which makes visible identification of the source difficult. I guess it might have been discussed when planning the campaigns. How did you overcome this problem? I am also curious to the similar mapping plots for $\delta^{13}C$ of $CH_4$. Spatial variability of $\delta^{13}C$ corresponding to those in the $CH_4$ mole fraction in Figure 2 is also valuable information for readers especially who consider plan of similar type of field surveys.

P11 L316: While the nighttime campaign detected more spikes with high $CH_4$ mole fractions, there are several peaks with comparable magnitude of $CH_4$ spikes (> 10 ppm) even in the daytime campaign. How are they explained? It would be helpful if you could add labels for key sources in Figure 1. I tried to compare Figs 1 and 2, but found it difficult to find corresponding locations of the peaks exactly.

P12 Section 3.2: Before going into results of source signatures, it would be good to show some selected examples of the observed Miller-Tans (or Keeling) plots from both campaigns. It could be done as Figure 3, not appendix figures. Such figures could represent how closely the observed $\delta^{13}C$ varied with the observed $CH_4$ mole fraction, which is key information that support validity of the characterization analysis for source isotope signatures presented in this study. Also from this point of view, I would prefer Keeling plot if possible, where magnitude of the observed $\delta^{13}C$ variations is obvious on the vertical axis. I am curious to how largely $\delta^{13}C$ varied in the plume air of different origins.

P12 L319: What is the "uncertainties" of the estimated source signatures? Please describe explicitly. It could be presented in section 2.4.

P12 L321: If so, it could be possible to see a characteristic feature when the estimated uncertainties are plotted versus range (or maximum) of the observed $CH_4$ mole fractions for individual plumes. Perhaps the authors could infer how much elevation in $CH_4$ mole fraction is desired for precise estimation of source isotope signature based on plume surveys like this study.

P13 Table 2: A number for each plume could be assigned. It would help readers to find a line in the Table corresponding to texts in the following sections.

P17 L424: "calm to light wind conditions" What is the wind speed at the observation time? There are similar subjective expressions also at other places. It is considered that degree of accumulation of $CH_4$ emitted from a nearby source is dependent on wind speed as the authors also explains in this manuscript. For example, constant 5 m/s wind brings influence of 300 m upwind source with a 1 minute delay, and we would collect footprint of 18 km upwind in an hour. It is therefore important to present an exact number to convince predominance of a nearby source. If no on-site measurement is available, the authors might look for data from a nearby weather station.

P18 L441: I am confused by this sentence. I understand that this study aims at characterizing isotope signatures of known sources, but here it seems the author tries to infer contributing sources based on isotope signature.

P18 L446: As far as I understand, a single source with $\delta^{13}C$ signature of -44.3±0.3‰ is not identified but interpreted as mixture of several potential sources. It is not clear that why the source signature is compared to those of a single source from different regions. Given that every isotope signature of the four potential sources is unknown, it is also difficult to infer possible contributions.

P19 L467: Same comment as that for P18 L441. The authors try identification of sources or separate contributions of sources, but it contrasts to this study's purpose of characterization of individual sources. I was therefore confused by impression that the objective change from section to section.

P19 L479ff: Chang et al. (2019) suggested that $\delta^{13}C_{CH4}$ from ruminants correlates with $\delta^{13}C$ of the diet as follows: $\delta^{13}C_{CH4} = 0.91 \times \delta^{13}C_{diet} - 43.49$. Could it be possible to check whether the present results are consistent with this equation? If the above equation is roughly valid for this study, $\delta^{13}C$ signature of the diet would be around -20‰, which lies between $\delta^{13}C$ of C3 and C4 plants. Is any information on the $\delta^{13}C$ of the diet available?

Chang et al. (2019) Revisiting enteric methane emissions from domestic ruminants and their $\delta^{13}C_{CH4}$ source signature, Nature Communications, https://doi.org/10.1038/s41467-019-11066-3.

P20 Section 3.2.6: What is the likely production process of $CH_4$ in the piggery?

P21 L539: According to Figure 6, ruminants in Australia show biomodal $\delta^{13}C$ signature distribution, while samples collected for cattle in this study show values at depleted side only.

P21 Section 3.3: When we consider use of $\delta^{13}C$ and $\delta D$ of $CH_4$ as constraints to the regional (e.g. Surat Basin) budget of $CH_4$, critical is the representativeness of individual source isotope signature. In this regard, it is good that some types of the sources in the region (e.g. Ground and river seeps, abattoir) showed good agreement within narrow isotope signature ranges between campaigns in different years, suggesting that source isotope signatures vary a little e.g. well representative. In contrast, some sources showed larger differences between campaigns or locations, suggesting that source isotope signatures could vary with time and/or space. It is therefore still uncertain that how representative the source isotope signatures presented in this study are at regional scale. To overcome this issue, one needs more frequent and numerous surveys, which would be highly challenging. Otherwise, a bit more zoomed-out scale study so that one can capture outflow of mixture air from the entire source unit (not source point to point) might help. I would like to suggest the authors to add discussions on how representative the source signatures presented in this study could be considered and on possible future sampling strategies to better comprehend isotope signatures of various sources in the region.

**Technical Comments**

P2 L43: "insights" to "data". If the authors have something else, please clarify.

P2 L48: "coal seam gas" to "CSG".

P4 L108: "n.d." to "2016"

P4 L130: "coal seam gas" to "CSG".

P5 L133: "chemistry" to "signature".

P6 L152: "coal seam gas" to "CSG"

P8 L216: "coal seam gas" to "CSG"

P9 L243: "closely" to "well"

P9 L252: "coal seam gas" to "CSG"

P11 L310: "light" to "weak"

P23 L594: "coal seam gas" to "CSG"

P24 L614: "coal seam gas" to "CSG"

---

## Author Comment (AC1)

Atmos. Chem. Phys. Discuss., referee comment RC1
https://doi.org/10.5194/acp-2021-76-RC1, 2021 ©
Author(s) 2021. This work is distributed under the
Creative Commons Attribution 4.0 License.

[Figure]

**Reply by the Authors to "Comment on acp-2021-76"**

Anonymous Referee #1
* * *
Referee comment on "Isotopic Signatures of Major Methane Sources in the Coal Seam Gas Fields and Adjacent Agricultural Districts, Queensland, Australia" by Xinyi Lu et al., Atmos. Chem. Phys. Discuss., https://doi.org/10.5194/acp-2021-76-RC1, 2021
* * *
My comments are attached as a supplement PDF file.

Please also note the supplement to this comment:
https://acp.copernicus.org/preprints/acp-2021-76/acp-2021-76-RC1-supplement.pdf

Review of the manuscript "**Isotopic Signatures of Major Methane Sources in the Coal Seam Gas Fields and Adjacent Agricultural Districts, Queensland, Australia**" by Xinyi Lu et al.

**Short summary of the manuscript**

This paper presents results from campaigns of in-situ $CH_4$ mole fraction measurements and of air samplings for subsequent isotope analyses. The authors characterized carbon and hydrogen stable isotope signatures of various $CH_4$ sources in the Surat Basin, Australia and compared the values with those in previously reported literature.

**General comment**

This is a good piece of work showing 2-year campaign measurement results. Previously, Sherwood et al. (2017) compiled thousands of isotope signature data of various $CH_4$ sources, but availability of such data is still limited for some regions or source types. This study will be acknowledged for complement of the available dataset and for important information for isotope based top-down estimate of the Australian $CH_4$ emissions. This work is well within the

scope of the journal and has significance in the study field. For publication of this manuscript, I would like to encourage the authors major revisions to consider my comments below.

Authors' response:

We thank the reviewer for placing the scientific value of this research in both the global and national context, and for the supportive comments and encouragement. Below we have addressed the constructive comments about the manuscript and highlighted where we have made revisions.

**1. Representativeness of the isotope signatures presented in this study**

This study presents isotope signatures of various sources in the Surat Basin. The results showed that the estimated isotope signatures of same source type sometimes differ from site to site. In such field studies, critically important is how representative the result is. Previous studies have inferred that $\delta^{13}C$ and $\delta D$ signatures of $CH_4$ emitted from some sources vary considerably with time and space. I am very curious to discussion on how representative the isotope signatures obtained in this study are. Such discussion is crucial when the results are considered for constraints to the regional $CH_4$ budget. There are two points I can come up with. First, whether the set of air samples were collected from downwind of the major emission locations of the source. The higher fraction of emission at the location among the total emission is, the more representative the results would be. Second, how large variability among isotope signatures from a single source type is. For instance, the results indicate that isotope signatures of $CH_4$ of CSG origin vary by 5 ‰ in $\delta^{13}C$ and by 20 ‰ in $\delta D$, depending on sampling location/time. But the variability is much smaller for ground seeps in $\delta^{13}C$, which may suggest the source isotope signatures are relatively constant/uniform. Such discussion could be made not only from the results from the present study but also from those from the earlier studies (Table 1). I would like to see more enriched discussions about what we can learn about representativeness from this study and give some suggestions about better sampling strategy for a similar study in future. It could affect discussions in section 3.3, because distinctiveness of isotope signatures of different source types depends on this issue. For instance, some part of $\delta^{13}C$ and $\delta D$ signatures overlap between fossil fuel and biogenic sources according to the Sherwood et al. (2017) database, but their global representative values are considered to be sufficiently different so that we can examine partition estimates for contributions of both sources to atmospheric $CH_4$ variations.

Authors' response:

We acknowledge the importance and challenges with quantifying how representative the presented isotopic signatures are.

With regards to "First, whether the set of air samples were collected from downwind of the major emission locations of the source.": we collected all air samples downwind of each source. A portion of the samples were collected close to centre line, and then additional samples towards the edge of the plume in order to maximise the spread of $CH_4$ mole fraction readings used in the Keeling regressions.

Regarding "Second, how large variability among isotope signatures from a single source type is.": the authors acknowledge that this is a challenge for studies of this nature. It would take considerable time and resources to collect multiple bag sets at multiple common sources. For this study we collected and analysed over 160 bag samples and

attempted to characterise the isotopic signature of all major sources in the region. For export abattoirs (meat works) and piggeries, these are the first isotopic signatures reported for these sources in Australia, and for most other sources these are the first deuterium measurements.

We have modified the following sentences in the manuscript:

P23 L594 – L602: "[…]  We present the $\delta^{13}C_{CH4}$ isotopic signatures for 16 plumes and the $\delta D_{CH4}$ isotopic signatures for 13 plumes, from the analyses of over 160 air samples. Despite the size of the data set, for many sources only a single isotopic signature has been determined. However, this single isotopic value represents the first recorded isotopic signature for some sources (e.g., abattoirs and piggeries) in Australia. Generally, the $\delta^{13}C_{CH4}$ and $\delta D_{CH4}$ signatures determined from isolated plumes mapped during our 2018 and 2019 campaigns agree with values reported in the literature (Table 1 and Fig. 6).  More investigations in Australia are needed for further characterisation of other sources, both those listed in the UNFCCC inventory classifications and natural. There is also a need for further studies to characterise the temporal and spatial variability of all sources, climatic and seasonal influences, and procedural repeatability. Ideally, further sampling should be undertaken in collaboration with the operators of each facility, so that samples can be collected closer to the source, removing all uncertainty in the origin of the CH₄. This study has made a contribution to the $\delta^{13}C_{CH4}$ and $\delta D_{CH4}$ signatures from different sources in Australia and internationally […]"

**2. Data analysis**

The authors employ the Miller-Tans plot. I could not however understand advantage of the analysis over the traditional Keeling plot. Keeling plot assumes that the background atmosphere is constant over the time period of interest, and thus provides a much simpler framework and data interpretation. In contrast, as shown by Miller and Tans (2003), Miller-Tans plot is useful when one needs to assume that the background atmospheric condition varies with time at significant level of magnitude. To my understanding, Miller-Tans plot should be chosen only for special cases, where constant background cannot be assumed. From this point of view, the authors argument "The background air normally does not change during this period but using the Miller-Tans approach is a safeguard against any variability." (P11 L295) seems strange to me.

Not only for the original Keeling (1958) paper (cited in the manuscript), the authors might revisit Miller and Tans (2003) for different ways of use of the two plots. Pataki et al. (2003) might also help for limitation of the Keeling plot.
In the case where the authors continue use of the Miller-Tans plot, they should clearly present how the time-varying background mole fractions and delta values are given for individual data points. It is not presented explicitly in the current manuscript. I suppose that it would for instance call for on-site continuous measurements in the upwind. Miller and Tans (2003) applied curvefitting to their time-series $CO_2$ data, and similar examples for CH₄ were given by Umezawa et al. (2012).

Additional useful reference is Zobits et al. (2006), who investigated difference of source isotope signatures between Keeling and Miller-Tans plots and between ordinary square and geometric mean regressions. They found that, for same conditions, both plots return practically identical source isotope signatures. However, when the range of mole fraction is relatively small, geometric mean regression method could cause a bias in the estimated source signature. Since this study targets relatively wide range of mole fraction variation, I suppose such a bias is small, but the authors could examine with their data and/or justify validity of their regression approach. According to the text, the "Bayesian regression" used in this study seems to be similar to geometric mean regression that could cause a bias for small range of mole fraction. Please explain more to justify the analysis method.

In summary, there would be no difference in estimated source signatures from both plots if constant background is assumed. However, unless the reason why Miller-Tans plot is needed is clearly given, I see that the authors make the analytical methodology complicated without necessity. And it is not logically clear why they chose the regression method. I would like to suggest the authors to revisit previous studies carefully and rewrite the corresponding section.

Pataki, D. E. et al. (2003), The application and interpretation of Keeling plots in terrestrial carbon cycle research. Global Biogeochem. Cycles, 17(1), 1022, https://doi.org/10.1029/2001GB001850.

Umezawa, T. et al. (2012), Contributions of natural and anthropogenic sources to atmospheric methane variations over western Siberia estimated from its carbon and hydrogen isotopes, Global Biogeochem. Cycles, 26, GB4009, https://doi.org/10.1029/2011GB004232.

Zobits, J. M. et al. (2006), Sensitivity analysis and quantification of uncertainty for isotopic mixing relationships in carbon cycle research. Agricultural and Forest Meteorology, 136, 56–75, https://doi.org/10.1016/j.agrformet.2006.01.003.

Authors' response:

We acknowledge that we were not precise with our statement in P11 L295. We agree that using the Keeling plot method improves the clarity of the manuscript and we now use this method throughout the manuscript. Converting to the Keeling plot method only results in minor changes (many only in the third significant figure) to the determined isotopic values and does not alter any of the discussion or conclusions in the originally submitted manuscript. We have updated the sentences in Sect. 2.4 P11 L287 – L299, P12 L318 – L322 and substitute the Keeling plots for Miller-Tans plots for all sampled plumes in the Appendix A (Figs. A3–A7).

P11 L287 – L299: "The $\delta^{13}C_{CH4}$ and $\delta D_{CH4}$ for $CH_4$ sources of each detected plume were determined using the  Keeling plot approach ( Keeling, 1958; Pataki et al., 2003) shown in Eq. (1):

$$\delta_{(a)} = \left[CH_{4(b)}\right](\delta_{(b)} - \delta_{(s)}) * 1/\left[CH_{4(a)}\right] + \delta_{(s)} \qquad (1)$$

where $[CH_{4(b)}]$ and $\delta_{(b)}$ are the $CH_4$ mole fraction and $\delta^{13}C_{CH4}$ (or $\delta D_{CH4}$) of the background air, $[CH_{4(a)}]$ and $\delta_{(a)}$ are the $CH_4$ mole fraction and $\delta^{13}C_{CH4}$ (or $\delta D_{CH4}$) of the atmosphere and $\delta_{(s)}$ is the $\delta^{13}C_{CH4}$ (or $\delta D_{CH4}$) of the mean source, respectively. The  intercept ($\delta_{(s)}$) of the linear regression between $\delta_{(a)}$  and $1/[CH_{4(a)}]$ represents the isotopic signature of the source mixed in the background ambient air. The Keeling plot method requires the background air $CH_4$ mole fraction and isotopic signature to be

constant during the period of observation. The time it takes to collect the 10 samples is approximately 30 minutes, and normally the background air composition does not change during  the period of sampling.  The mobile survey readings show that the background $CH_4$ mole fraction was stable in 2018 and 2019 daytime and nighttime surveys (Fig. A2), which supports this assumption. For each  Keeling data set the Bayesian linear regression line and credible interval (analogous to confidence interval) were determined using the PyMC3 Bayesian regression package (Salvatier et al., 2016). The regression methodology was selected based on the fact that there are bivariant correlated errors in both the x and y variables (e.g., Miller and Tans, 2003; Zazzeri et al., 2016) and the number of samples in each plume set was small (<= 10). Bayesian regression was used since it is a robust algorithm that balances uncertainty in both the x and y axis data (Jaynes and Crow, 1999),  it is suitable for small data sets (Baldwin and Larson, 2017), and it has been demonstrated to yield more reliable isotopic signatures at low mole fractions with low sample numbers (Zobitz et al., 2007).

P12 L318 – L322: "The Keeling plot results of $CH_4$ source signature calculations are listed in Table 2 and shown in Fig. 3. The  Keeling plots are shown in Figs. A3–A7 in Appendix A.  For each $\delta^{13}C_{CH4}$ (‰) and $\delta D_{CH4}$ (‰) isotopic signature both the posterior standard deviation and the  credible interval  were determined. The variability in the  credible interval is primarily due to both the  sampled $CH_4$ mole fraction range and the number of data points used in the  Keeling plot analysis as shown in Fig. A8. "

We have also updated the caption of Table 2 as follows:

P13 L329: "Table 2: $CH_4$ source signature results for plumes sampled in the Surat Basin 2018 and 2019 campaigns. $CH_4$ excess over background (ppm) for the samples that were used to calculate the source signature. $\delta^{13}C_{CH4}$ (‰) and $\delta D_{CH4}$ (‰) are reported along with the Bayesian posterior distribution mean, standard deviation and 95 % credible interval (in brackets). NA: not applicable."

Within the bounds of the credible interval (the Bayesian measure of uncertainty) the derived isotopic signatures are similar, and the overall interpretation of the results does not alter. Throughout the text we have replaced "uncertainty" with the Bayesian statistical terminology "credible interval" to convey the use of Bayesian statistics for all regression analyses.

The determination of the line of best fit for the Keeling plots is a generic regression problem. It is well established in the statistical analysis literature that when there is error in both the x and y variables the use of ordinary least squares is not appropriate. This is discussed in Miller and Tans (2003). The authors have used in numerous publications the bivariant correlated error with scatter (BCES) algorithm (Akritas and Bershady, 1996); for example, refer to the cited ACP publication Zarreri et al. (2016). On the GitHub web resources page for the BCES python scripts there is a discussion on the advantages of Bayesian linear regression over the BCES algorithm. https://github.com/rsnemmen/BCES.

The reviewer has pointed to the study of Zobitz et al. (2006) for justification to use ordinary least squares regression. We would like to highlight that for a small sample size Keeling plot, Zobitz et al. (2007) used Bayesian regression (Fig. 2 in that paper) and demonstrated that it performed better than ordinary least squares regression for small data sets.

Bayesian linear regression is now commonly used in many fields of science where there is a need to characterise parameter uncertainty for regression of small sample size data sets, when there is the likelihood of outliers and there is error in both the x and y variables. The authors feel it is beyond the scope of this manuscript to do a regression methodology comparison study for fitting Keeling and Miller–Tans regression lines, especially when Bayesian regression is now commonly used, is available in many mathematical packages, and there are both R and Python libraries on GitHub.

For this manuscript we have stayed with using Bayesian regression, especially since Zobitz et al. (2007) established that Bayesian regression is a robust algorithm for Keeling plot analyses. We also note that there is no significant difference between the isotopic signatures determined for all samples using either the Keeling or Miller–Tans methods.

**3. Description of the instrument performance**

The authors should present their performance tests of the instruments (Los Gatos and Picarro) made before the campaigns. It seems that the current descriptions rely on information provided by the product companies only, which is performance of their standard products at shipment from factory, but we know that there are differences among products and importantly that stability of those instruments (e.g. during shipment, repetitive power on/off, change in temperature/humidity) has not been well established. It is important to show readers sufficient key information on validity of their measurements, independent from that from the provider.

Authors' response:

In the manuscript text below and Appendix A (Fig. A1) we now provide the infield performance data for both the UNSW Sydney LGR-UGGA and Picarro G2201-i CRDS units. The Picarro G2201-i CRDS and LGR-UGGA units were used only to locate the plumes, and we used the Picarro G2201-i CRDS $CH_4$ mole fraction data for checking if the bags leaked when the samples were shipped between UNSW Sydney and RHUL. None of the Keeling plot results use the UNSW Sydney Picarro data. We presented only a low-resolution image of the plume positions, to convey the scale of the project, the frequency of major plumes, and the order of magnitude of enhancement in the $CH_4$ mole fraction readings at the sampling locations. We acknowledge that changes in temperature and humidity affect both concentration and isotopic measurements using laser spectrometers. It is for this same reason that data used for Keeling plot analysis was obtained under laboratory conditions using a well-established analytical technique (IRMS).

We also added the following text in the manuscript:

P9 L245: "[…] The in-field standard deviations for mean $CH_4$ mole fraction measurements of the reference standard across all days were 4.9 ppb (2018) and 9.6 ppb (2019) for

LGR-UGGA and 5.3 ppb (2018) for Picarro G2201-i CRDS. This repeatability is better than reported in Takriti et al. (2021)."

P9 L254: "[...] purposes. In 2018 the root-mean-square error (RMSE) between the University of New South Wales (UNSW Sydney) Picarro 2201-i CRDS and the RHUL Picarro G1301 CRDS (detailed below) was 0.437 (ppm; Fig. A1 (a)) and in 2019 the RMSE between the UNSW Sydney Picarro 2201-i CRDS and the Institute for Marine and Atmospheric research Utrecht (IMAU) continuous-flow isotope ratio mass spectrometry (CF-IRMS) (detailed below) was 0.232 (ppm; Fig. A1 (b))

The following reference is added:

Takriti, M., Wynn, P. M., Elias, D. M. O., Ward, S. E., Oakley, S. and McNamara, N. P.: Mobile methane measurements: Effects of instrument specifications on data interpretation, reproducibility, and isotopic precision, Atmos. Environ., 246, 118067, https://doi.org/10.1016/j.atmosenv.2020.118067, 2021.

**Specific Comments**

P1 L14: "The use of…" This sentence needs rewriting. It is not clear how the authors consider difference of the three verbs. I think that (use of) $\delta^{13}C$ and $\delta D$ can "help us" identify a specific source "if potential sources are all characterized in $\delta^{13}C$ and $\delta D$ signatures."

Authors' response:

We thank the Referee for this suggestion. To clarify the wording the sentence has been rephrased:

P1 L14: "The characterisation of carbon ($\delta^{13}C$) and hydrogen ($\delta D$) stable isotopic composition of $CH_4$ can help distinguish between specific emissions of $CH_4$ […].

P1 L16: It is not clear how different "discriminate" is from for instance "distinguish between" in the earlier sentence.

Authors' response:

To clarify the wording the sentence has been rephrased:

P1 L16: "[…] This research examines whether dual isotopic signatures of $CH_4$ can be used to distinguish between sources of $CH_4$ in the Surat Basin."

P2 L40: "e.g." should be added before "Nisbet et al. 2020", otherwise additional reference that cover "greenhouse gases" is needed. I think the reference is for $CH_4$ only.

Authors' response:

Agreed and revised to the following:

P2 L40: "to anthropogenic industrial and agricultural activities (e.g., Nisbet et al., 2020) […]"

P2 L60: The CH$_4$ increase over the industrial era and stagnation during the early 2000s are phenomena with totally different time scales.

Authors' response: Agreed and revised to the following:

P2 L60 – L61: "[…]  The CH$_4$ mole fraction has increased by 160 % since industrialisation. The rate of increase is typically 0.4 to 14.7 ppb per year, although there was a short pause in the growth rate of atmospheric CH$_4$ between 1999 and 2006 (Schaefer et al., 2016; Dlugokencky, 2021) […]"

The following reference is added:

Dlugokencky, E.J.: Annual Increase in Globally-Averaged Atmospheric Methane, NOAA/GML, https://www.esrl.noaa.gov/gmd/ccgg/trends_ch4/, last access: 21 April 2021.

P3 L65: There are indeed many references that addressed different time phases of the atmospheric CH$_4$ increase. First, I would like to suggest the authors to cite references that substantially contributed to the present study only. Did all these references contributed equally to the present study? Second, as written in the sentence, such debate has continued over the last decades, while the references are all relatively new. It might give wrong impression that the problem is new. Some "old" but key references, for instance Steele et al. (1992), Dlugokencky et al. (1998), Bousquet al. (2006), Simpson et al. (2012) and others, could be considered. Additionally, a recent study Chandra et al. (2021) present conclusion similar to that by Jackson et al. (2020). That said, choice of references is up to the authors.

Steele, L., Dlugokencky, E., Lang, P. et al. Slowing down of the global accumulation of atmospheric methane during the 1980s. Nature 358, 313–316 (1992). https://doi.org/10.1038/358313a0.

Dlugokencky, E., Masarie, K., Lang, P. et al. Continuing decline in the growth rate of the atmospheric methane burden. Nature 393, 447–450 (1998). https://doi.org/10.1038/30934.

Bousquet, P., Ciais, P., Miller, J. et al. Contribution of anthropogenic and natural sources to atmospheric methane variability. Nature 443, 439–443 (2006). https://doi.org/10.1038/nature05132.

Simpson, I., Sulbaek Andersen, M., Meinardi, S. et al. Long-term decline of global atmospheric ethane concentrations and implications for methane. Nature 488, 490–494 (2012). https://doi.org/10.1038/nature11342.

Chandra, N., P. K. Patra, J. S. H. Bisht, et al. Emissions from the Oil and Gas Sectors, Coal Mining and Ruminant Farming Drive Methane Growth over the Past Three Decades. Journal of the Meteorological Society of Japan (2021). https://doi.org/10.2151/jmsj.2021015.

Authors' response:

We agree with the Referee's comment. Indeed, some key references in the past have substantially contributed to the current study of $CH_4$ increase. We have therefore made the following changes:

P3 L65 – L67: "[…] between agriculture versus fossil fuels (Bousquet et al., 2006; Chandra et al., 2021;  Jackson et al., 2020; Kirschke et al., 2013; Nisbet et al., 2014, 2016, 2019; Rice et al., 2016; Rigby et al., 2017;  Schwietzke et al., 2016; Turner et al., 2017; Worden et al., 2017) […]"

The following references will be added to the updated manuscript:

Bousquet, P., Ciais, P., Miller, J. B., Dlugokencky, E. J., Hauglustaine, D. A., Prigent, C., Van Der Werf, G. R., Peylin, P., Brunke, E. G., Carouge, C., Langenfelds, R. L., Lathière, J., Papa, F., Ramonet, M., Schmidt, M., Steele, L. P., Tyler, S. C. and White, J.: Contribution of anthropogenic and natural sources to atmospheric methane variability, Nature, 443(7110), 439–443, https://doi.org/10.1038/nature05132, 2006.

Chandra, N., Patra, P. K., Bisht, J. S. H., Ito, A., Umezawa, T., Saigusa, N., Morimoto, S., Aoki, S., Janssens-Maenhout, G., Fujita, R., Takigawa, M., Watanabe, S., Saitoh, N. and Canadell, J. G.: Emissions from the Oil and Gas Sectors, Coal Mining and Ruminant Farming Drive Methane Growth over the Past Three Decades, J. Meteorol. Soc. Japan. Ser. II, 2021–015, https://doi.org/10.2151/jmsj.2021-015, 2021.

The following reference will be deleted from the updated manuscript:

P3 L69: "natural fossil fuel source" looks strange. All fossil fuels are of natural origin in nature. The difference is just emission takes place by nature or by human. Etiope and colleagues have used the term "geological". Please rephrase.

Authors' response: Agreed and revised to the following:

P3 L69: "although this result contradicts emission estimates on the size of  geological  $CH_4$ sources (Etiope et al., 2019) […]"

P3 L69: What is the current global fraction of emissions from unconventional sources in the total fossil fuel related $CH_4$ emissions?

Authors' response:

The authors agree that information on the current global fraction of emissions from unconventional sources in the total fossil fuel related $CH_4$ emissions is important. To help the reader to understand this issue, a sentence was added to P3 L74:

P3 L74: "CH$_4$ emissions (Lan et al., 2019). It is estimated that around 14 % of total fossil fuel CH$_4$ emissions are from unconventional sources in 2020 (IEA, 2021) […]"

The following reference will be added to the updated manuscript:

IEA: Methane Tracker Database, IEA, Paris: https://www.iea.org/articles/methane-tracker-database, last access: 8 April 2021, 2021.

P4 L121: Same comment as that for P3 L65.

Authors' response:

We have made the following changes:

P4 L120 – L122: "(Beck et al., 2012; Fisher et al., 2017; France et al., 2016; Lowry et al., 2020; McNorton et al., 2018; Nisbet et al., 2016, 2019; Rice et al., 2016; Rigby et al., 2017; Röckmann et al., 2016; ; Schwietzke et al., 2014, 2016; Tarasova et al., 2006; ) […]"

P8 L219: What is the actual precision evaluated by the study team and expected during the campaign? Please see my earlier comment.

Authors' response:

We thank the Referee for the comment. Please refer to our response for General comment 3.

P9 L225: The same question as that for P8 L219. The information here seems to be identical to those on the Data Sheet provided on the Picarro website

(https://www.picarro.com/support/library/documents/g2201i_analyzer_datasheet). These are general instrument performance when it is shipped from factory. I suppose that the study team did not use the instrument as delivered but carried out series of evaluations before the campaign. The authors should present those results that better represent the actual performance during the campaign.

Authors' response:

We thank the Referee for the comment. Please refer to our response for General comment 3.

P9 L231: The more details of the time lag correction should be presented. How did you estimate the time lag? Is it time dependent or constant? How long is it on average?

Authors' response:

We have made the following changes:

P9 L231 – L233: "[…] ~~To correct for the time lag between GPS location and CRDS recorded data caused by slow flow rate and inlet tube length (~ 2.5 m), we adjusted the time stamp of CH$_4$ mole fraction and δ$^{13}$C$_{CH4}$ readings based on observed delay of the~~

 Using the standard air, we determined the time lag between the real-time GPS location reading and the display of mole fraction reading on the Picarro G2201-i CRDS to be 3 min and 40 s. Using this timing offset, we adjusted the time stamp for the analyser data."

P9 L243: Please describe the value explicitly. What is the exact value and uncertainty measured by another laboratory? I believe the measurement was made by INSTAAR, not NOAA/GML.

Authors' response:

Our intention was to demonstrate that the RHUL $\delta^{13}C_{CH4}$ measurement of the calibration air provided by CSIRO was in good agreement with the flasks collected at Cape Grim around the same time. To clarify the wording the sentence has been rephrased:

P9 L241 – L244: "[…] RHUL also measured the $CH_4$ mole fraction of the calibration gas (1801.2 ± 0.5 ppb),  The isotope value measured by RHUL (−47.2 ± 0.05 ‰) also closely resembles the value from flasks (−47.2 ± 0.04 ‰, mean ± standard deviation for 12 flasks collected) collected at Cape Grim and measured at the  Institute of Arctic and Alpine Research (INSTAAR), University of Colorado (White et al., 2018) […]"

P9 L247: Was the driving speed adjusted specifically for the different instruments (e.g. flow rate)? If so, additional information will help readers.

Author's response: Please refer to the response to General Comment 3.

P10 L267: Here the authors refer to the WMO X2004A scale traceable by the gas cylinders provided by NOAA. In the section before, they wrote "WMO scale" only. Reading these sentences, I cannot be sure that the mole fraction values from CSIRO and RHUL are on the identical scale. Please sort this out so that use of one single measurement scale is clear throughout the paper. Though the principal methodology is still valid, since the original WMO X2004 scale was updated to WMO X2004A in 2015, the reference Dlugokencky et al. (2005) no longer represents the current WMO scale exactly. I would suggest to additionally cite the NOAA website (https://www.esrl.noaa.gov/gmd/ccl/ch4_scale.html) or the latest WMO GAW Report No. 255 (https://library.wmo.int/index.php?lvl=notice_display&id=21758#.YE7BLS9h0UE). It should be somewhere noted that the current WMO scale cover up to ~5900 ppb and many measurements presented in this study are calculated by extrapolation of the scale. The abbreviation "NOAA" appeared earlier.

Authors' response:

We apologise for the unclear description of the calibration scale. The above-mentioned calibration air (appeared in P9 L235 – L239) was provided by CSIRO but measured by RHUL to link analysers to the same WMO X2004A scale of that at RHUL. We have made the following changes for clarification:

P9 L236 – L239: "by Commonwealth Scientific and Industrial Research Organisation (CSIRO)  The calibration gas was  placed into 3 litre SKC FlexFoil PLUS sample bags (SKC Inc., USA) for shipping and analysed at the greenhouse gas laboratory of Royal Holloway, University of London (RHUL) to determine the $\delta^{13}C_{CH4}$ for the calibration air (−47.2 ± 0.05‰). RHUL also measured the $CH_4$ mole fraction of the calibration gas (1801.2 ± 0.5 ppb).[…]"

P10 L267 – L268: "calibrated to the WMO X2004A scale using NOAA (National Oceanic and Atmospheric Administration) air standards (Dlugokencky et al., 2005; Fisher et al., 2006, 2011; WMO 2020) […]"

The following reference will be added to the updated manuscript:

WMO: 20th WMO/IAEA Meeting on Carbon Dioxide, Other Greenhouse Gases and Related Measurement Techniques (GGMT-2019), Jeju Island, South Korea, 2–5 September 2019, GAW Report No. 255, 140 pp., https://library.wmo.int/inde x.php?lvl=notice_display&id=21758#.YJzyYmYzbUI, 2020.

P10 L276: I would suggest a sentence like "For the subsequent IRMS measurement, $CH_4$ in sample air in most bags were preconcentrated for 10 minutes at…, but that in samples with $CH_4$ mole fraction larger than 6 ppm reported by RHUL were processed for shorter time in order to…" if I understand correctly. I suppose that "sampled" in the original sentence is not just collecting air but preconcentration of $CH_4$ from sample air, as written in the following sentence.

Authors' response:

That is correct. To clarify we have rephrased the sentence to the following:

P10 L276: "[…]  For the subsequent IRMS measurement, the $CH_4$ in air from most bags were preconcentrated for 10 minutes at a flow rate of 6 mL min⁻¹ for $\delta D_{CH4}$ and 4 mL min⁻¹ for $\delta^{13}C_{CH4}$, but for samples reported by RHUL that had a $CH_4$ mole fraction larger than 6 ppm they were  processed for a shorter time in order to extract a quantity of $CH_4$ similar to the reference air.

P10 L283: What does "UNSW" stands for?

Authors' response:

UNSW is the abbreviation for The University of New South Wales, Sydney, Australia. The official abbreviation for the university is UNSW Sydney. We have defined the abbreviation in the author listing and change UNSW to "UNSW Sydney" throughout the text.

P11 Section 2.4: Please see my earlier comment.

Authors' response: Please refer to our response for General comment 2.

P11 Section 3.1: More than half of sentences of this paragraph are for explaining the switch from daytime to nighttime sampling. I do not think the title of the section represent the content well. A problem of nighttime samplings might be darkness, which makes visible identification of the source difficult. I guess it might have been discussed when planning the campaigns. How did you overcome this problem? I am also curious to the similar mapping plots for $\delta^{13}C$ of $CH_4$. Spatial variability of $\delta^{13}C$ corresponding to those in the $CH_4$ mole fraction in Fig. 2 is also valuable information for readers especially who consider plan of similar type of field surveys.

Authors' response:

The authors agree that a more representative title is needed. We have made changes as follows:

P11 L301: "**3.1 Regional plume mapping and the benefits of sampling at nighttime**"

P11 L306: "In 2018, we did not detect plumes from coal mines, river seeps, abattoirs, piggeries or WWTP, thus we shifted our focus from daytime surveying […]"

Regarding the problem locating sources when doing nighttime sampling, most facilities are well lit. We added the following sentence:

P11 L313: "plots and minimises the uncertainties of the derived isotopic source signatures. As part of developing an inventory (Neininger et al., in review) in the region, all major $CH_4$ sources were located and were georeferenced to guide nighttime sampling. Also, most facilities were well lit, which assisted with source identification. The contrast in the magnitude of the $CH_4$ mole fraction […]"

The following reference has been added:

Neininger, B.G., Kelly, B.F.J., Hacker, J.M., Lu, X., Schwietzke, S., Coal seam gas industry methane emissions in the Surat Basin, Australia: Comparing airborne measurements with inventories, Phil. Trans. R. Soc. A., in review.

Regarding "similar mapping plots for $\delta^{13}C$ of $CH_4$": it was never the goal to produce isotope maps, the goal was to find the $CH_4$ plumes. We used the Picarro G2201-i CRDS for only a small portion of the survey when the LGR-UGGA failed. Only a small portion of the complete mole fraction survey would have isotope data from Picarro G2201-i CRDS, which is not sufficient to produce a similar map.

P11 L316: While the nighttime campaign detected more spikes with high $CH_4$ mole fractions, there are several peaks with comparable magnitude of $CH_4$ spikes (> 10 ppm) even in the daytime campaign. How are they explained? It would be helpful if you could add labels for key sources in Figure 1. I tried to compare Figs 1 and 2 but found it difficult to find corresponding locations of the peaks exactly.

Authors' response:

We cannot provide a detailed answer because we did not analyse the flux of each source. The $CH_4$ mole fraction at the point of sampling is a complex function of source strength, distance to the source, wind speed, wind direction, surface roughness, temperature, uplift rate, among other variables. We have placed source numbers on Fig. 1 in line with Table 2 and modified the caption of Fig. 1 as follows:

P8 L213: "Queensland (Inset map data: Australian Government (2020) ), Administrative Boundaries © Geoscape Australia). The positions of the sampled $CH_4$ plumes are numbered 1 through 16."

P12 Section 3.2: Before going into results of source signatures, it would be good to show some selected examples of the observed Miller-Tans (or Keeling) plots from both campaigns. It could be done as Figure 3, not appendix figures. Such figures could represent how closely the observed $\delta^{13}C$ varied with the observed $CH_4$ mole fraction, which is key information that support validity of the characterization analysis for source isotope signatures presented in this study. Also, from this point of view, I would prefer Keeling plot if possible, where magnitude of the observed $\delta^{13}C$ variations is obvious on the vertical axis. I am curious to how largely $\delta^{13}C$ varied in the plume air of different origins.

Authors' response:

We believe that the required details are succinctly summarised for all sources in Table 2 to inform a reader on "how largely $\delta^{13}C$ varied in the plume air of different origins". In updated Table 2 all derived $\delta^{13}C$ and $\delta D$ values are reported, along with the credible interval now added for those parameters.

Keeling plots have also been provided now in the Appendix A (Figs. A3–A7) and cross referenced in the caption of Table 2.

P12 L319: What is the "uncertainties" of the estimated source signatures? Please describe explicitly. It could be presented in section 2.4.

Authors' response: Please refer to our response to General comment 2.

P12 L321: If so, it could be possible to see a characteristic feature when the estimated uncertainties are plotted versus range (or maximum) of the observed $CH_4$ mole fractions for individual plumes. Perhaps the authors could infer how much elevation in $CH_4$ mole fraction is desired for precise estimation of source isotope signature based on plume surveys like this study.

Authors' response:

We thank the reviewer for this suggestion. The precise estimation of source isotope signatures depends on several factors such as maximum $CH_4$ mole fractions observed, number of samples collected, distance from source and wind direction. We have presented the dependency between 95 % credible interval range of $\delta^{13}C$ (a) and $\delta D$ (b) derived from Keeling plot method and number of samples and measured $CH_4$ mole

fraction range from the corresponding measured sources in Fig. A8 in Appendix A. The result agrees well with previous studies (e.g., Hoheisel et al., 2019; Takriti et al., 2021).

With respect to the measurement systems, we have added the following sentence:

P10 L285: "[…] Due to the high precision of the RHUL GC-IRMS measurements of < 0.05 ‰ for $\delta^{13}C$, the IMAU IRMS measurements of < 0.1 ‰ for $\delta^{13}C$ and < 2 ‰ for $\delta D$, reliable source signatures can usually be derived for elevations of 100–200 ppb above the background"

P13 Table 2: A number for each plume could be assigned. It would help readers to find a line in the Table corresponding to texts in the following sections.

Authors' response:

We thank the Referee for making this use recommendation. We have assigned a number for each plume in Table 2 and added the plume number when referencing the plume in the manuscript.

P17 L424: "calm to light wind conditions" What is the wind speed at the observation time? There are similar subjective expressions also at other places. It is considered that degree of accumulation of $CH_4$ emitted from a nearby source is dependent on wind speed as the authors also explains in this manuscript. For example, constant 5 m/s wind brings influence of 300 m upwind source with a 1 minute delay, and we would collect footprint of 18 km upwind in an hour. It is therefore important to present an exact number to convince predominance of a nearby source. If no on-site measurement is available, the authors might look for data from a nearby weather station.

Authors' response:

Calm to light wind descriptions was the official Australian Bureau of Meteorology wind speed description at the time of sampling – 0 to 14 $km^{-1}$.

The speed has been added in brackets after the description in the manuscript:

P17 L424: "plumes near the Chinchilla weir and measured $CH_4$ mole fractions as high as 18 ppm in calm to light wind conditions (0–14 km $h^{-1}$) […]"

P18 L441: I am confused by this sentence. I understand that this study aims at characterizing isotope signatures of known sources, but here it seems the author tries to infer contributing sources based on isotope signature.

Authors' response:

The $CH_4$ isotope signatures of Abattoir A and B were calculated to represent the abattoir facilities instead of a single source. We have updated in the manuscript to highlight that the $CH_4$ isotope signatures presented in this study are either for a single source (e.g., venting pipeline, raw water pond) or a facility with several potential $CH_4$ sources (e.g., abattoirs, feedlots, and piggeries).

We have also made the following changes to P18 L441:

P18 L441 – L442: "integrated feedlot and processing plant. "

P18 L446: As far as I understand, a single source with $\delta^{13}C$ signature of -44.3±0.3 ‰ is not identified but interpreted as mixture of several potential sources. It is not clear that why the source signature is compared to those of a single source from different regions. Given that every isotope signature of the four potential sources is unknown, it is also difficult to infer possible contributions.

Authors' response:

We agree that the potential sources of the mixed urban emissions are unknown, and it is not suitable to be compared with other single sources or single facilities. We therefore removed the Mixed urban emissions from Table 2 and Fig. 3, deleted Sect. 3.2.9 and the plots in Appendix and the following sentence in the manuscript:

P7 L193 – L196: "~~Each town centre has many potential sources of CH₄ including, but not limited to, leaking gas bottles, instant hot water systems, rubbish bins, vehicles and domestic wood fires (which are common in the region). To characterise these collective emissions, samples were collected from a typical residential area in Dalby, which has a population of approximately 12,000 (Australian Bureau of Statistics, 2016).~~"

P12 L321 – L322: ""

P19 L467: Same comment as that for P18 L441. The authors try identification of sources or separate contributions of sources, but it contrasts to this study's purpose of characterization of individual sources. I was therefore confused by impression that the objective change from section to section.

Authors' response: Please see the response to P18 L441.

P19 L479ff: Chang et al. (2019) suggested that $\delta^{13}C_{CH4}$ from ruminants correlates with $\delta^{13}C$ of the diet as follows: $\delta^{13}C_{CH4} = 0.91 \times \delta^{13}C_{diet} - 43.49$. Could it be possible to check whether the present results are consistent with this equation? If the above equation is roughly valid for this study, $\delta^{13}C$ signature of the diet would be around -20‰, which lies between $\delta^{13}C$ of C3 and C4 plants. Is any information on the $\delta^{13}C$ of the diet available?

Chang et al. (2019) Revisiting enteric methane emissions from domestic ruminants and their $\delta^{13}C_{CH4}$ source signature, Nature Communications, https://doi.org/10.1038/s41467-01911066-3.

Authors' response:

We do not have the required diet information to do this calculation. No edit was made.

P20 Section 3.2.6: What is the likely production process of CH₄ in the piggery?

Authors' response:

The possible production process of CH$_4$ in the piggery was introduced in P7 L177 – L180.

P21 L539: According to Figure 6, ruminants in Australia show biomodal δ$^{13}$C signature distribution, while samples collected for cattle in this study show values at depleted side only.

Authors' response:

The δ$^{13}$C$_{CH4}$ signature of cattle emitted CH$_4$ varies depending on many factors such as geographical location, diet (C3 or C4 plant) and breed of the animal (Hook et al., 2010). In Australia, cattle are hosted in different states with different geographical locations thus have varying diets. These factors could possibly explain the bimodal δ$^{13}$C$_{CH4}$ signature distribution from limited data available. Further studies are also important to better understand the δ$^{13}$C$_{CH4}$ signature of cattle emitted CH$_4$ from cattle in Australia. No edit was made.

P21 Section 3.3: When we consider use of δ$^{13}$C and δD of CH$_4$ as constraints to the regional (e.g. Surat Basin) budget of CH$_4$, critical is the representativeness of individual source isotope signature. In this regard, it is good that some types of the sources in the region (e.g. Ground and river seeps, abattoir) showed good agreement within narrow isotope signature ranges between campaigns in different years, suggesting that source isotope signatures vary a little e.g. well representative. In contrast, some sources showed larger differences between campaigns or locations, suggesting that source isotope signatures could vary with time and/or space. It is therefore still uncertain that how representative the source isotope signatures presented in this study are at regional scale. To overcome this issue, one needs more frequent and numerous surveys, which would be highly challenging. Otherwise, a bit more zoomed-out scale study so that one can capture outflow of mixture air from the entire source unit (not source point to point) might help. I would like to suggest the authors to add discussions on how representative the source signatures presented in this study could be considered and on possible future sampling strategies to better comprehend isotope signatures of various sources in the region.

Authors' response: Please see the response to General Comment 1.

**Technical Comments**

P2 L43: "insights" to "data". If the authors have something else, please clarify.

Authors' response: Agreed and revised to the following:

P2 L43: "in conjunction with other data. While ethane measurements have been used […]"

P2 L48: "coal seam gas" to "CSG".

Authors' response: Agreed and revised to the following:

P2 L48: "largest CSG fields is co-located with large scale cattle feedlots […]"

P4 L108: "n.d." to "2016"

    Authors' response: Agreed and revised to the following:

    P4 L108: "[…] because it is often co-emitted in fossil fuel emissions (Conley et al., 2016; Dlugokencky et al., 2011; Lowry et al., 2020; Smith et al., 2015) […]"

P4 L130: "coal seam gas" to "CSG".

    Authors' response: Agreed and revised to the following:

    P4 L130: "Here we present mobile $CH_4$ surveys in the CSG fields in southeast […]"

P5 L133: "chemistry" to "signature".

    Authors' response: Agreed and revised to the following:

    P5 L133: "database on the isotopic signature of $CH_4$ sources in Australia […]"

P6 L152: "coal seam gas" to "CSG"

    Authors' response: Agreed and revised to the following:

    P6 L152: "[…] All the CSG in the Surat Basin […]"

P8 L216: "coal seam gas" to "CSG"

    Authors' response: Agreed and revised to the following:

    P8 L216: "the main roads throughout the major CSG and agricultural regions of the Surat Basin […]"

P9 L243: "closely" to "well"

    Authors' response: Agreed and revised to the following:

    P9 L243: "agrees well with the value from CSIRO […]"

P9 L252: "coal seam gas" to "CSG"

    Authors' response: Agreed and revised to the following:

    P9 L252: "samples were collected from 16 major sources in the Surat Basin CSG fields […]"

P11 L310: "light" to "weak"

    Authors' response: Agreed and revised to the following:

    P11 L310: "[…] at night during weak to moderate wind conditions […]"

P23 L594: "coal seam gas" to "CSG"

Authors' response: Agreed and revised to the following:

P23 L594: "eastern Surat Basin CSG fields in Queensland, Australia […]"

P24 L614: "coal seam gas" to "CSG"

Authors' response: Agreed and revised to the following:

P24 L614: "signatures from CSG sources overlap with signatures expected from landfills […]"

**References**

Akritas, M. G. and Bershady, M. A.: Linear Regression for Astronomical Data with Measurement Errors and Intrinsic Scatter, Astrophys. J., 470, 706, https://doi.org/10.1086/177901, 1996.

Hoheisel, A., Yeman, C., Dinger, F., Eckhardt, H. and Schmidt, M.: An improved method for mobile characterisation of $\delta^{13}CH_4$ source signatures and its application in Germany, Atmos. Meas. Tech., 12(2), 1123–1139, https://doi.org/10.5194/amt-12-1123-2019, 2019.

Hook, S. E., Wright, A. D. G. and McBride, B. W.: Methanogens: Methane producers of the rumen and mitigation strategies, Archaea, 2010, https://doi.org/10.1155/2010/945785, 2010.

Miller, J. B. and Tans, P. P.: Calculating isotopic fractionation from atmospheric measurements at various scales, Tellus B, 55(2), 207–214, https://doi.org/10.1034/j.1600-0889.2003.00020.x, 2003.

Takriti, M., Wynn, P. M., Elias, D. M. O., Ward, S. E., Oakley, S. and McNamara, N. P.: Mobile methane measurements: Effects of instrument specifications on data interpretation, reproducibility, and isotopic precision, Atmos. Environ., 246, 118067, https://doi.org/10.1016/j.atmosenv.2020.118067, 2021.

Zazzeri, G., Lowry, D., Fisher, R. E., France, J. L., Lanoisellé, M., Kelly, B. F. J., Necki, J. M., Iverach, C. P., Ginty, E., Zimnoch, M., Jasek, A. and Nisbet, E. G.: Carbon isotopic signature of coal-derived methane emissions to the atmosphere: from coalification to alteration, Atmos. Chem. Phys, 16, 13669–13680, https://doi.org/10.5194/acp-16-13669-2016, 2016.

Zobitz, J. M., Burns, S. P., Ogée, J., Reichstein, M. and Bowling, D. R.: Partitioning net ecosystem exchange of CO2: A comparison of a Bayesian/isotope approach to environmental regression methods, J. Geophys. Res. Biogeosciences, 112(G3), https://doi.org/10.1029/2006JG000282, 2007.

---

## Author Comment (AC2)

Atmos. Chem. Phys. Discuss., referee comment RC1
https://doi.org/10.5194/acp-2021-76-RC1, 2021 ©
Author(s) 2021. This work is distributed under the
Creative Commons Attribution 4.0 License.

[Figure]

**Reply by the Authors to "Comment on acp-2021-76"**

Anonymous Referee #2
* * *
Referee comment on "Isotopic Signatures of Major Methane Sources in the Coal Seam Gas Fields and Adjacent Agricultural Districts, Queensland, Australia" by Xinyi Lu et al., Atmos. Chem. Phys. Discuss., https://doi.org/10.5194/acp-2021-76-RC2, 2021
* * *
General comments:
In this manuscript, Lu et al. present mobile atmospheric methane measurements conducted in 2018 and 2019 and subsequent laboratory analyse of the $\delta C$ and $\delta D$ in $CH_4$ signature of 17 methane sources in the Surat Basin. During the campaigns, mobile GHG analysers are used to identify emission plumes of different important $CH_4$ sources and whole-air samples taken within those plumes are later analysed in the lab by suitable means. The results are then compared with previous studies from Australia and globally. The authors also highlight the added value of double isotope analysis, i.e., $\delta D$ and $\delta C$ in $CH_4$.

Overall, the manuscript is well written and nicely structured, which makes it easy to follow. The issue of source apportionment of anthropogenic and natural methane sources continuous to be of great importance in this field of research with possible future policy implications. This study reflects important incremental progress and the fact that it emphasis the value of double isotope analysis make it very relevant beyond the immediate study region. The quality and scope of the paper is, therefore, fully suitable for publication in Atmospheric Chemistry and Physics. However, there are two general issues and a few very minor comments that should be addressed before publication.

Author's response:

We thank the reviewer for this constructive feedback on the manuscript, and the positive comment about this research being important. Below we address the concerns raised, and we have followed the suggestions where we feel that they improve the quality of the manuscript. All authors thank the reviewer for reading the manuscript carefully, and for making insightful recommendations.

■ Representativeness of results:

Without further information on the type of landfill and WWTP for example it seems impossible to judge if the found isotopic signatures can be used for other such facilities in the regions. The same problem arises for the piggery and especially the abattoirs where the source mix/methane producing process seems unclear. Are the isotopic signatures presented here representative for each class of facility in the region or will future studies have to be conducted to characterize each piggery and abattoir seen in Figure 1? This should be discussed in more detail in the conclusion section.

> Author's response:
>
> We acknowledge the importance and challenges with quantifying how representative the presented isotopic signatures are. Collating statistically robust data sets for the isotopic signature of all primary $CH_4$ sources is a time-consuming and costly endeavour. For this study we collected and analysed over 160 bag samples. Some of the data presented in the manuscript are the first and only data we have for that source under Australian conditions (e.g., abattoirs and piggeries).
>
> We have modified the following sentences in the manuscript:
>
> P23 L594 – L602: "[…] $_{CH4}$$_{CH4}$ We present the $\delta^{13}C_{CH4}$ isotopic signatures for 16 plumes and the δD isotopic signatures for 13 plumes, from the analyses of over 160 air samples. Despite the size of the data set, for many sources only a single isotopic signature has been determined. However, this single isotopic value represents the first recorded isotopic signature for some sources (e.g., abattoirs and piggeries) in Australia. Generally, the $\delta^{13}C_{CH4}$ and $\delta D_{CH4}$ signatures determined from isolated plumes mapped during our 2018 and 2019 campaigns agree with values reported in the literature (Table 1 and Fig. 6). $_{CH4}$ More investigations in Australia are needed for further characterisation of other sources, both those listed in the UNFCCC inventory classifications and natural. There is also a need for further studies to characterise the temporal and spatial variability of all sources, climatic and seasonal influences, and procedural repeatability. Ideally, further sampling should be undertaken in collaboration with the operators of each facility, so that samples can be collected closer to the source, removing all uncertainty in the origin of the $CH_4$. This study has made a contribution to the $\delta^{13}C_{CH4}$ and $\delta D_{CH4}$ signatures from different sources in Australia and internationally […]"

■ Calculation of uncertainties:

The uncertainties reported seem to only rely on the uncertainty of the Miller-Tans fit. However, for some plumes e.g. the river seep (Figure A2) only 4 data points are available and any sub-sample of 3 of those data points could yield different results. As the isotopic signatures rely on very few data points per site (some on only 3 data points) it seems reasonable to use leave-one-out-validation to check if the Miller-Tans fit uncertainty is reasonable. For some sites the current uncertainties could just reflect the lower bound of the true uncertainty, which is not highlighted in the conclusion section.

Author's response:

We acknowledge that the small sample size is an issue with some of the bag sets collected to determine the isotopic signature of individual sources. "Leave-one-out" or "Bootstrapping" is not appropriate for such small data sets, because there are too few points to establish the population distribution statistics from the limited number of permutations. We used Bayesian regression, because it is widely considered as one of the more robust measures for characterising parameter uncertainty, via the credible interval (analogous to the confidence interval in ordinary least squares regression). Please refer to Beckman and Cook (1983) and McNeish (2016).

Beckman, R. J. and Cook, R. D.: Outlier … … …. s, Technometrics, 25(2), 119–149, https://doi.org/10.1080/00401706.1983.10487840, 1983.

McNeish, D.: On Using Bayesian Methods to Address Small Sample Problems, Struct. Equ. Model. A Multidiscip. J., 23(5), 750–773, https://doi.org/10.1080/10705511.2016.1186549, 2016.

The Bayesian Regression workflow and determination of the credible interval is comprehensively discussed on the Wolfram web pages:

https://blog.wolfram.com/2019/08/22/embracing-uncertainty-better-model-selection-with-bayesian-linear-regression/.

There is also a good introductory discussion on the difference between Bayesian and Ordinary Least Squares regression by Koehrsen (2018):

https://towardsdatascience.com/introduction-to-bayesian-linear-regression-e66e60791ea7

We have replaced the term "uncertainty (ies)" with "credible interval" to better convey the use of Bayesian statistics, not frequentist statistics.

Specific and technical comments:

L5: If appropriate, consider spelling out the name of the University of New South Wales Sydney here.

Author's response:

UNSW Sydney is the official brand of our institution. We have made the following change:

L5: "[…] The University of New South Wales Sydney (UNSW Sydney) […]"

L17: Why is this limited to 'warm and hot climate regions''? Even in temperate and colder climates, nighttime GHG levels are known to be enhanced compared to afternoon values due to less vertical mixing in a lower PBL.

Author's response:

The authors agree; therefore, the sentence has been rephrased to the following:

L17: "[…] We also highlight the benefits of sampling at nighttime  […]"

L19: Were all 17 plumes analysed for $\delta D_{CH4}$? Only 14 sites are given in Figure 3 (and the tables) what happened to the $\delta D_{CH4}$ data from the other sites (e.g. the Chinchilla landfill)?

Authors' response:

We acknowledge that this sentence about the $\delta D_{CH4}$ analyses was not clear. Unfortunately, due to a limited budget, only a selected set of samples was analysed for $\delta D_{CH4}$. To clarify, we have added "from 13 sources" in L273:

L273: "[…] A portion of the samples (from 13 plumes) was further analysed in the Institute for Marine and Atmospheric research"

L40: Nisbet et al. 2020 is a great reference here, but IPCC AR5 (you cite Myrhe et al. 2013 later on) and others (e.g. Ganesan et al. 2019) could be/should be mentioned as well.

https://agupubs.onlinelibrary.wiley.com/doi/full/10.1029/2018GB006065

Authors' response:

We thank the reviewer for this suggestion, and we have added these most useful references (in addition to corrections implemented in RC#1 – Comment P2 L40): L40 now reads "[…] to anthropogenic industrial and agricultural activities (e.g. Ganesan et al., 2019; IPCC, 2014; Nisbet et al., 2020)."

The following references are added to the updated manuscript:

Ganesan, A. L., Schwietzke, S., Poulter, B., Arnold, T., Lan, X., Rigby, M., Vogel, F. R., van der Werf, G. R., Janssens-Maenhout, G., Boesch, H., Pandey, S., Manning, A. J., Jackson, R. B., Nisbet, E. G. and Manning, M. R.: Advancing Scientific Understanding of the Global Methane Budget in Support of the Paris Agreement, Global Biogeochem. Cycles, 33(12), 1475–1512, https://doi.org/10.1029/2018GB006065, 2019.

Pachauri, R. K. , Allen, M. R. , Barros, V. R. , Broome, J. , Cramer, W. , Christ, R. , Church, J. A. , Clarke, L. , Dahe, Q. , Dasgupta, P. , Dubash, N. K. , Edenhofer, O. , Elgizouli, I. , Field, C. B. , Forster, P. , Friedlingstein, P. , Fuglestvedt, J. , Gomez-Echeverri, L. , Hallegatte, S. , Hegerl, G. , Howden, M. , Jiang, K. , Jimenez Cisneroz, B. , Kattsov, V. , Lee, H. , Mach, K. J. , Marotzke, J. , Mastrandrea, M. D. , Meyer, L. , Minx, J. , Mulugetta, Y. , O'Brien, K. , Oppenheimer, M. , Pereira, J. J. , Pichs-Madruga, R. , Plattner, G. K. , Pörtner, H. O. , Power, S. B. , Preston, B. , Ravindranath, N. H. , Reisinger, A. , Riahi, K. , Rusticucci, M. , Scholes, R. , Seyboth, K. , Sokona, Y. , Stavins, R. , Stocker, T. F. , Tschakert, P. , van Vuuren, D. and van Ypserle, J. P., Pachauri R.K. and Meyer L.A. (Eds.): Climate Change 2014: Synthesis Report. Contribution of Working Groups I, II and III to the Fifth Assessment Report of the Intergovernmental Panel on Climate Change, IPCC, Geneva, Switzerland, 151 pp., https://epic.awi.de/id/eprint/37530/, 2014.

L41: Suggest rephrasing to "not **always** possible" or "not **easily** possible" as it is indeed possible to identify sources from atmospheric mole fraction measurements when sources are **not** juxtaposed and isolated enough and/or when suitable calibrated atmospheric modelling can be performed.

Authors' response:

The authors thank the reviewer for this suggestion. We have edited L41 to read:

L41: "in the atmosphere it is not always possible to isolate the source of the emission, especially if many sources are juxtaposed […]"

L152 (and following): Why is 'coal seam gas' not abbreviated as CSG here? Suggest to consistently use CSG throughout the manuscript. Possible exception at beginning of new sections if readers 'jump' directly to the conclusions it might be useful to reintroduce CSG.

Authors' response:

Agreed and revised throughout the text. Please also refer to our response to RC#1 – Technical Comments regarding the consistent use of the abbreviation CSG.

L157: Further information on the power stations seems necessary, how many are gasfired and how many are coal-fired? Also, they account for 4.7% of GHG emissions from the electricity sector, but is this mostly due to their $CO_2$ emissions or do they report significant loss rates for $CH_4$ as well?

Authors' response:

We thank the Referee for the comment. We've added the number of gas- and coal-fired power stations. There is now a comprehensive inventory for the region of study, including details on how the $CH_4$ emissions were estimated, in Neininger et al. (under review). The estimated $CH_4$ emissions from the power plants is 0.15 % of the regional $CH_4$ emissions. L157 has been edited to read:

L157: "[…] In the study area, seven power stations (5 CSG–fired and 2 coal–fired) are operational in the area of study and together they account for 0.15 % of the $CH_4$ emissions for the south-east portion of the Surat Basin CSG fields (Neininger et al., in review) […]"

The following reference has been added:

Neininger, B.G., Kelly, B.F.J., Hacker, J.M., Lu, X., Schwietzke, S. Coal seam gas industry methane emissions in the Surat Basin, Australia: Comparing airborne measurements with inventories, Phil. Trans. R. Soc. A., in review, 2021.

L174: Correct to "fa**r**ming"

Authors' response: Agreed and revised to the following:

L174: "[…] Most cattle in the region are in the surrounding dryland farming districts […]"

L199: Here the authors nicely outline which factors determine landfill emissions, but none of the suggested parameters controlling landfill $CH_4$ production is reported for the Chinchilla landfill.

Authors' response:

Little or no information is in the public domain for small country town landfills in Australia. Chinchilla has a population of 6612 people at the last census (https://quickstats.censusdata.abs.gov.au/census_services/getproduct/census/2016/quickstat/SSC30606).

We added the following sentence at L202:

L202: "[…] The landfill is typical of many small-town landfills in the region, and when operational it accepted mixed dry and solid organic domestic waste, commercial and industrial waste. These landfills have a simple design and typically have a clay lining and soil cover. A full listing of the landfills in the study area and the materials deposited within each are listed in Western Downs Regional Council (2021a)."

The following reference has been added:

Western Downs Regional Council: Waste Facilities & Disposal Fees, https://www.wdrc.qld.gov.au/living-here/environment-and-health/waste-disposal/waste-facilities/, last access: 29 April, 2021a.

L201: The amount of (organic) waste deposited at this landfill site would seem a critical parameter to add here (its disposal area might be of secondary importance).

Authors' response:

We agree with the Referee that knowing the amount of (organic) waste deposited at this landfill site would be beneficial. However, due to limited public information, no specific data about the proportion of organic versus other waste could be found. Please refer to the response to Comment L199.

L205: What kind of waste treatment is used at the Miles WWTP, e.g. are sludge digesters used?

Authors' response:

We thank the Referee for highlighting this. We have added more details as follows:

L205: "[…] In 2019 we sampled the plume immediately downwind of the Miles wastewater treatment plant. There, the sludge was treated in digestion tanks under anaerobic conditions. The liquid from the tanks was then transferred to the aerobic lagoons for further purifying (Western Downs Regional Council, 2021b)."

We added the following reference:

Western Downs Regional Council: Regional Sewerage Networks, https://www.wdrc.qld.gov.au/living-here/engineering-services/utility-services/wastewater-and-sewerage/regional-sewerage-networks/#miles-sewerage, 29 April, 2021b.

L220-L230: Suggestion to cite appropriate peer-reviewed studies on instrument performance rather than only manufacturer specifications. Previous studies have investigated the performance of G2201i and UGGA instruments e.g.

https://www.sciencedirect.com/science/article/pii/S0956053X17309698

https://www.gov.uk/government/uploads/system/uploads/attachment_data/file/331683/S_C130034_Report.pdf

https://amt.copernicus.org/articles/8/4539/2015/

https://amt.copernicus.org/articles/10/2077/2017/

Authors' response:

We agree that further details on the performance of the analysers should be added. It needs to be noted that only the Picarro G2201-i CRDS $CH_4$ mole fraction data from mobile surveys were used in this manuscript. All Keeling and Miller–Tans plot analyses used the results from the higher precision RHUL Picarro G1301 CRDS, GC-IRMS and IMAU CF-IRMS systems as detailed in the manuscript. We modified the following text and provided in-field performance data in the Appendix (please refer to our response to RC#1 – General Comment 3):

L219: "records the $CH_4$ mole fraction  every second in parts per million (ppm). The manufacturer's  stated precision is  of <2 parts per billion (ppb) and a measurement range of 0 to 100 ppm. These analysers were further characterised by Allen et al. (2019). In-field calibration using southern-ocean air supplied by CSIRO is discussed further below. The air inlet […]"

L230: "[…] of < 1.15 ‰ at 10 ppm. Previous studies have also characterised the Picarro G2201-i performance (e.g., Assan et al., 2017; Rella et al., 2015) […]"

The following references has been added:

Allen, G., Hollingsworth, P., Kabbabe, K., Pitt, J. R., Mead, M. I., Illingworth, S., Roberts, G., Bourn, M., Shallcross, D. E. and Percival, C. J.: The development and trial of an unmanned aerial system for the measurement of methane flux from landfill and greenhouse gas emission hotspots, Waste Manag., 87, 883–892, https://doi.org/10.1016/j.wasman.2017.12.024, 2019.

Assan, S., Baudic, A., Guemri, A., Ciais, P., Gros, V. and Vogel, F. R.: Characterization of interferences to in situ observations of $δ^{13}CH_4$ and $C_2H_6$ when using a cavity ring-down spectrometer at industrial sites, Atmos. Meas. Tech., 10(6), 2077–2091, https://doi.org/10.5194/amt-10-2077-2017, 2017.

Rella, C. W., Hoffnagle, J., He, Y. and Tajima, S.: Local- and regional-scale measurements of $CH_4$, $δ^{13}CH_4$, and $C_2H_6$ in the Uintah Basin using a mobile stable isotope analyzer, Atmos. Meas. Tech., 8(10), 4539–4559, https://doi.org/10.5194/amt-8-4539-2015, 2015.

L224: The cited precision of 8cm is likely only for static measurements. Typically, the limiting factor for the resolution on mobile platforms is the GPS frequency. Was the 10Hz or the optional 20Hz version of the A326 used here?

Authors' response:

We acknowledge that the precision of the GPS is for the static measurement. We have modified the following text:

L221 – L223: "[…] A Hemisphere GPS (Model A326, Hemisphere GNSS Inc., USA) was also mounted on the roof, which measures the static geolocation to within 8 cm (2 standard deviations, GNSS 2017). The air inlet tube was 2.5 m long; this results in a lag between the GPS recorded time stamp and the analyser time stamp. Using standard air this was determined to be 7 s. It was not the goal of the project to do detailed plume analyses. Driving speed was not independently continuously measured, and only a lag time correction was made. As a result, the surveys were not precisely positioned. When a major plume was traversed, we returned to the centreline of the plume and remained stationary to georeference the plumes shown in Fig. 2. The car was stationary for up to half an hour while the air samples were collected. In Fig. 2 the plume positions are accurately located, but away from the plumes the survey results are only approximate to within the order of tens of meters."

L230: see L224

Authors' response: Please refer to our response to Comment L224.

L319: Are those uncertainties only based on the uncertainty of the fit or was a bootstrapping method used as well to check that individual data points do not overly bias the slope? Was any data selection applied, i.e. only accepting slopes of fits with an $R^2$ above a threshold value?

Authors' response:

Bayesian regression is a robust regression method – it was selected because it reduced the influence of outliers on the derived statistics. The uncertainty, or better-termed credible interval, is not determined from a single line of best fit; rather it is determined from the posterior distribution of the parameters.

L325 – Figure 2: The three reference levels of $CH_4$ (3, 2 and 1.8ppm) are rather confusing and make the figure more difficult to follow. As all major peaks are labelled with their concentration this seems unnecessary (or maybe just have one reference level). The grey ribbon is hardly visible with blue and green levels on the figure.

Authors' response:

As recommended by the reviewer we have removed the 3 and 2 ppm reference level lines but left the reference 1.8 ppm line in the updated Fig. 2.

L330 – Table 1: Please add the range or maximum $CH_4$ mole fraction for each plume, additionally the Pearson's R or $R^2$ of the Miller-Tans fit might be informative. Furthermore, it seems important to highlight if the samples were part of a daytime or nighttime survey.

Authors' response:

R and $R^2$ are only suitable measures for ordinary least squares regression (a frequentist approach). There is no universally accepted Bayesian equivalent of R or $R^2$. Please refer to Gelman et al. (2019) for further details on this topic. The goal in using Bayesian regression is to robustly determine the credible interval for the isotopic signature. It is not to assess the deviation of points from the linear regression model, which is the information provided by R and $R^2$.

We have added to Table 2 the highest excess over background for the samples used to calculate the source signature.

L339 – Could ground migration/stray natural gas migration be important here? If yes, this means emissions can be significantly displaced from the actual leaking infrastructure.

Authors' response:

We believe that the ground migration/stray natural gas migration is not of concern here. The $CH_4$ enhancement measured in the study was closely related to the source (infrastructure) we identified. At all CSG locations, distinct plumes were discernible, and we traced each plume close to each facility.

L387 – Figure 4: The current figure makes it difficult to compare literature sources and this study. A two-panel figure with the same size and same X&Y scaling could make it easier. Alternatively, adding PM, SM and T areas to the main part of Figure 4 would achieve the same.

Authors' response:

We thank the reviewer for this feedback and have adopted the suggestion. We have changed Fig. 4 in the manuscript.

L395 – The abbreviation ROM seems not to be used elsewhere in the manuscript.

Authors' response:

We thank the Referee for highlighting this oversight, and we have made the following change in the sentence:

L395: "with permission to extract up to 2.8 million tonnes per annum (Mtpa) of run-of-mine  coal (Yancoal, 2018) […]"

L440 – Why are no $\delta D_{CH4}$ values reported for abattoir A?

Authors' response:

Unfortunately, due to a limited budget, only a selected set of samples was analysed for $\delta D_{CH4}$. Please refer to our response for Comment L19 for additional discussion on this point.

L469 – Please state more clearly what your theory is about the source of the CH$_4$ from abattoir A and B. There is a discussion of different options which are likely not the source and eventually, I think, you are suggesting that there is a biogas generator on site or a waste lagoon that is causing the emissions?

Also, if one of the abattoirs actually has an integrated feedlot and we cannot be sure what the source is, are the values found for this abattoir applicable to other abattoirs?

> Authors' response:
>
> There are only two large meat processing facilities in the region – Beef City and Oakey Beef Exports. Throughout the text when referring to these meatworks we now write abattoirs (meat works and processing) to separate them from licensed abattoir locations, which may be a small facility on a private property (such details are not provided with the government records). In Fig. 1, we now show them separately as "export abattoirs (meat works and processing)" and "licensed abattoirs".
>
> Both abattoirs are multi-purpose facilities, consisting of animal management areas, slaughter facilities with venting stacks, meat and other processing as stated in L446 to L448. Due to limited site access, we were unable to closely investigate individual sources. We therefore highlighted in L450 that the plume we sampled was most likely from mixed sources (a meat works facility value, not an isolated source value).

L523 – What kind of aerobic and/or anaerobic treatment is implemented at the Miles wastewater treatment plant? A simple description in section 2. might be helpful to understand why it is more similar to some WWTPs from previous studies.

> Authors' response: Please refer to our response to Comment L205.

L530: Was the consumer grade natural gas used in the city measured or were samples from wood stoves taken? If not, those seem important sources to add in future studies.

> Authors' response:
>
> We have deleted the contents about mixed urban emissions. Please refer to our response for RC#1 – Comment P18 L446. We did this for two reasons. At the regional level we have now determined that the CH$_4$ emissions from the towns is only a minor source of CH$_4$ (Neininger et al. (under review). Also, as these urban results were a highly blended signature, and the focus of the paper is on individual sources or facilities, we have removed all discussions on the urban emissions and blended isotopic signature from the manuscript. Ideally each source in a town should be characterised, but that was beyond the scope of this project.

L590-L600: Seems to be a summary, rather than contain conclusions. Other parts of this section also seem to repeat previous results. Therefore, renaming the section to "Summary and conclusion" might be appropriate.

> Authors' response:
>
> We agree with the Referee; the section has been renamed as follows:
>
> L592: "4 Summary"

L607: Are the samples for piggeries and abattoirs representative of other facilities of the same type in Queensland or even Australia (see general comments).

Authors' response:

The majority of piggeries in Australia have a common design, to maximise the health and safety of the pigs, and to reduce the environmental impacts. For example, refer to:

http://australianpork.com.au/wp-content/uploads/2018/08/NEGIP_2018_web.pdf

L625 – Suggest to add Pearson's R or $R^2$ for each Miller-Tans fit in Figures A1, A2. A3, A4 and A5.

Authors' response:

As discussed already, R and $R^2$ are frequentist statistical measures for ordinary least squares (OLS) regression. There is no direct equivalent measure for Bayesian regression. We did not use ordinary least squares regression, because it is well established that when there is error in both the X and the Y variables it is not an appropriate method to use.

**References**

Beckman, R. J. and Cook, R. D.: Outlier … … …. s, Technometrics, 25(2), 119–149, https://doi.org/10.1080/00401706.1983.10487840, 1983.

Gelman, A., Goodrich, B., Gabry, J. and Vehtari, A.: R-squared for Bayesian Regression Models, Am. Stat., 73(3), 307–309, https://doi.org/10.1080/00031305.2018.1549100, 2019.

McNeish, D.: On Using Bayesian Methods to Address Small Sample Problems, Struct. Equ. Model. A Multidiscip. J., 23(5), 750–773, https://doi.org/10.1080/10705511.2016.1186549, 2016.